# Human embryonic lung epithelial tips are multipotent progenitors that can be expanded in vitro as long-term self-renewing organoids

Marko Z Nikolić[1], Oriol Caritg[1], Quitz Jeng[1], Jo-Anne Johnson[1], Dawei Sun[1], Kate J Howell[2,3], Jane L Brady[1], Usua Laresgoiti[1], George Allen[1], Richard Butler[1], Matthias Zilbauer[2,4], Adam Giangreco[5†], Emma L Rawlins[1,6,7]*

[1]Wellcome Trust/CRUK Gurdon Institute, University of Cambridge, Cambridge, United Kingdom; [2]Department of Paediatrics, University of Cambridge, Cambridge, United Kingdom; [3]European Molecular Biology Laboratory, European Bioinformatics Institute, Wellcome Genome Campus, Cambridge, United Kingdom; [4]Department of Paediatric Gastroenterology, University of Cambridge and Addenbrookes Hospital, Cambridge, United Kingdom; [5]Lungs for Living Research Centre, UCL Respiratory, University College London, London, United Kingdom; [6]Department of Pathology, University of Cambridge, Cambridge, United Kingdom; [7]Wellcome Trust/MRC Stem Cell Institute, Cambridge, United Kingdom

*For correspondence: e.rawlins@gurdon.cam.ac.uk

Present address: †The Technology Partnership, Melbourn Science Park, Melbourn, United Kingdom

Competing interests: The authors declare that no competing interests exist.

**Abstract** The embryonic mouse lung is a widely used substitute for human lung development. For example, attempts to differentiate human pluripotent stem cells to lung epithelium rely on passing through progenitor states that have only been described in mouse. The tip epithelium of the branching mouse lung is a multipotent progenitor pool that self-renews and produces differentiating descendants. We hypothesized that the human distal tip epithelium is an analogous progenitor population and tested this by examining morphology, gene expression and in vitro self-renewal and differentiation capacity of human tips. These experiments confirm that human and mouse tips are analogous and identify signalling pathways that are sufficient for long-term self-renewal of human tips as differentiation-competent organoids. Moreover, we identify mouse-human differences, including markers that define progenitor states and signalling requirements for long-term self-renewal. Our organoid system provides a genetically-tractable tool that will allow these human-specific features of lung development to be investigated.

## Introduction

During mouse lung development the distal tip epithelial cells are $SOX9^+$, $ID2^+$ and function as multi-potent progenitors producing first bronchiolar, and then alveolar, descendants (*Alanis et al., 2014*; *Rawlins et al., 2009*). Between ~E10–15 cells that exit the distal tip turn off SOX9, upregulate SOX2 and differentiate along bronchiolar lineages. Whereas, ~E16–18 cells exiting the tip turn off SOX9 and co-express markers of alveolar type 1 (AT1) and alveolar type 2 (AT2) fate. As morphogenesis proceeds these bipotent cells line developing alveolar sacs and differentiate as mature AT1 or AT2 cells (*Desai et al., 2014*; *Treutlein et al., 2014*). Many factors controlling self-renewal and differentiation in the developing mouse lung epithelium have been identified. By contrast, relatively little is known about human lung development. This is largely due to practical considerations about tissue availability and culture system limitations (*Benlhabib et al., 2015*; *Haitchi et al., 2009*;

**eLife digest** Degenerative lung disease occurs when the structure of the lungs breaks down, which makes it harder to get enough oxygen into the bloodstream. Most, but not all, cases occur in smokers and ex-smokers or people who have been exposed to a lot of air pollution. Currently, there is no way to reverse the damage, and even slowing the progress of the disease is extremely difficult. Some researchers are looking for ways to treat patients with degenerative lung diseases by regenerating the surface of their lungs. However, it is still not clear what the most effective route towards this long-term goal will be.

One approach to lung regeneration is to use findings from developmental biology to understand how embryos normally build the gas exchange surfaces in the lungs. This knowledge may allow scientists to trigger a similar process in an adult lung to renew or replace any diseased tissue. Alternatively, cells could be collected from patients, reprogrammed and then coaxed into becoming a gas exchange surface in the laboratory. Such a "lung-in-a-dish" could be used to understand how degenerative diseases develop, to discover and test new drugs, or even to treat the patient directly via a transplant.

To date, the embryonic development of lungs has mostly been studied using mouse lungs as a model system. However, it was not clear if human lungs actually develop in similar ways to mouse lungs, and whether using mice is a valid research strategy.

Nikolić et al. compared embryonic lungs from humans and mice and showed that they are indeed very similar in terms of the cell types that they contain and how they mature. However, some key differences were identified that can only be explored in human cells and tissue. Nikolić et al. went on to identify conditions that allowed them to grow cells from human embryonic lungs indefinitely in a dish. These cells can now be used to investigate the aspects of lung development that are specific to humans.

Together these findings provide a useful guide to allow scientists to coax human cells growing in a laboratory to become lung cells. Further improvements to this process will make the lungs-in-a-dish more true to the real organs, meaning that they could be used to better understand lung disease and identify new medicines. In the longer term, Nikolić et al. hope to gain enough insight from the human lung-in-a-dish model to eventually be able to regenerate the lungs of patients with degenerative lung disease. However, this possibility is still many years away.

Rajatapiti et al., 2010). A small number of human studies show the detailed expression of specific genes (Al Alam et al., 2015; Gonzalez et al., 1996; Khoor et al., 1993, Khoor et al., 1994; Laresgoiti et al., 2016; Stahlman et al., 2007; Zhang et al., 2012). Whereas, transcriptomics has provided a genome-wide view of human lung developmental transitions, but currently lacks cellular resolution (Feng et al., 2014; Kho et al., 2010, 2016).

Improved in vitro models of human disease are needed to complement available mouse models. One recent approach to disease modelling is to use self-renewing human organoids which recapitulate aspects of organ morphogenesis/physiology (Dekkers et al., 2013; Ettayebi et al., 2016; Huch et al., 2015). Human organoids are typically derived from adult stem cells limiting their use for studying paediatric disease and disease progression. An alternative is to derive the organ of interest from pluripotent stem cells by directed differentiation (Dye et al., 2016; Merkle and Eggan, 2013). The ability to in vitro self-renew and differentiate bona fide human lung tip progenitors could provide a genetic system for fundamental developmental biology and paediatric disease modelling. Moreover, an improved understanding of human lung progenitor states and human fate specification would facilitate strategies for directed differentiation of pluripotent cells.

We have extensively characterized human epithelial tip progenitors, and the early stages of fate specification, revealing mouse-human differences in the expression of key marker genes including SOX2 and pro-SFTPC. We developed methods to grow human tip epithelium as long-term self-renewing, branching, organoids and therefore investigate tip signalling requirements and differentiation capacity. Our human tip organoids can be directed to differentiate towards alveolar or bronchiolar fate in vitro and can engraft into the adult or developing mouse lung. These experiments

confirm that human and mouse tips are analogous in function and validate our organoid conditions. These conditions are sufficient to convert differentiating human embryonic lung epithelial stalks to tip fate, illustrating the plasticity of the developing lung. However, they do not support the long-term self-renewal of mouse tips, highlighting species-specific regulatory differences. Our organoid system thus provides a genetically-tractable in vitro model to accelerate studies of human lung development.

## Results

### Initiation of alveolar and bronchiolar differentiation in human embryonic lungs

We characterized the evolution of marker gene expression in human embryonic lungs 6–21 pcw (post-conception weeks; *Figure 1—figure supplement 1*). This period covers the pseudoglandular stage (~5–16 pcw) when the bronchiolar tree is established and the early canalicular stage (~16–21 pcw) when alveolar sac formation begins (*Burri, 1984*; *Rackley and Stripp, 2012*). Throughout the time-course, we observed SOX9 (SRY-box 9) localized to the distal epithelial tips (*Figure 1A,B*). In the adult human lung, pro-SFTPC (Surfactant Protein C, or SPC) and a monoclonal antibody, HTII-280, are markers of AT2 cells (*Figure 1C*) (*Barkauskas et al., 2013*; *Gonzalez et al., 2010*). We first detected low levels of HTII-280 in SOX9⁻ epithelium at 11 pcw, adjacent to the distal tip (*Figure 1A*). Similarly at 14 and 17 pcw (*Figure 1—figure supplement 2A*). At 20 pcw, HTII-280 was more heterogeneous in the alveolar sacs with cells having either high, or undetectable, levels (*Figure 1B*). We confirmed that HTII-280 was not in the larger airways (*Figure 1B*, arrowhead), but it was ubiquitous in the columnar epithelium of the terminal bronchioles (*Figure 1B*, arrow). By contrast, pro-SFTPC was first detected at very low levels at 17 pcw, particularly in tip epithelium (*Figure 1—figure supplement 3D*), and more robustly in distal squamous cells by 20 pcw (*Figure 1D–F*). At 20 pcw, pro-SFTPC was mostly co-expressed with HTII-280 (*Figure 1F'*), although we also observed a small number of pro-SFTPC⁺, HTII-280⁻ cells and many more pro-SFTPC⁻, HTII-280⁺ cells. This human embryonic pro-SFTPC staining differs from that observed in mouse development in which pro-SFTPC is expressed throughout the epithelium from ~E10, increases in canalicular stage tips and is further up-regulated in differentiating AT2 cells (*Laresgoiti et al., 2016*; *Wuenschell et al., 1996*). However, our human data are consistent with previous reports of pro-SFTPC staining in human embryos (*Khoor et al., 1994*). We previously used LPCAT1 (Lysophosphatidylcholine acyltransferase 1) as a specific marker of alveolar/AT2 fate in mouse development (*Laresgoiti et al., 2016*). However, it was expressed widely throughout the epithelium of the developing human lung from six pcw, although specific to AT2 cells in the adult (*Figure 1—figure supplement 2E,F*). ABCA3 (ATP binding cassette subfamily A member three) is present on the surface of lamellar bodies in mature AT2 cells, but has not been detected in human embryonic lungs prior to 28 weeks gestation (*Stahlman et al., 2007*). Consistent with this, we could not detect ABCA3 in any of the embryonic lungs we stained (6–21 pcw), but reproducibly saw expression in adult AT2 cells (*Figure 1—figure supplement 2F*). This suggests that mature lamellar body-containing AT2 cells are not present in human embryonic lungs by 21 pcw consistent with previous analysis (*Oulton et al., 1980*).

In mouse development the AT1 marker HOPX (Hop Homeobox) is first detected ubiquitously in cells that have exited the canalicular stage distal tip, subsequently becoming heterogeneous and marking future AT1 cells (*Jain et al., 2015*; *Laresgoiti et al., 2016*). Similarly, we detected HOPX in cells that had exited the distal tip at 11 pcw, although there was also co-expression of HOPX and SOX2 (SRY-Box 2) in the smaller airways (*Figure 1G*). HOPX and HTII-280 were frequently co-expressed between 11 and 17 pcw (*Figure 1—figure supplement 2B*). By 20 pcw, a mixture of single and co-expressing distal epithelial cells were observed lining the developing alveolar sacs (*Figure 1H*). Interestingly, at 20 pcw cells in the terminal bronchioles co-expressed SOX2, HTII-280 and HOPX (*Figure 1H'* and *Figure 1—figure supplement 2C*). A monoclonal antibody, HTI-56, has been reported as human AT1 cell specific (*Dobbs et al., 1999*; *Gonzales et al., 2015*). Consistent with this, we observed the highest levels of expression in HOPX⁺ cells adjacent to the distal tip from 11 pcw (*Figure 1I*). By 20 pcw HTI-56 was becoming heterogeneous, whilst HOPX remained ubiquitous (*Figure 1J,J'*). However, we also observed a low level of HTI-56 expression throughout the SOX2⁺ airways, beyond the terminal bronchioles, making it less useful as an AT1 marker in vitro

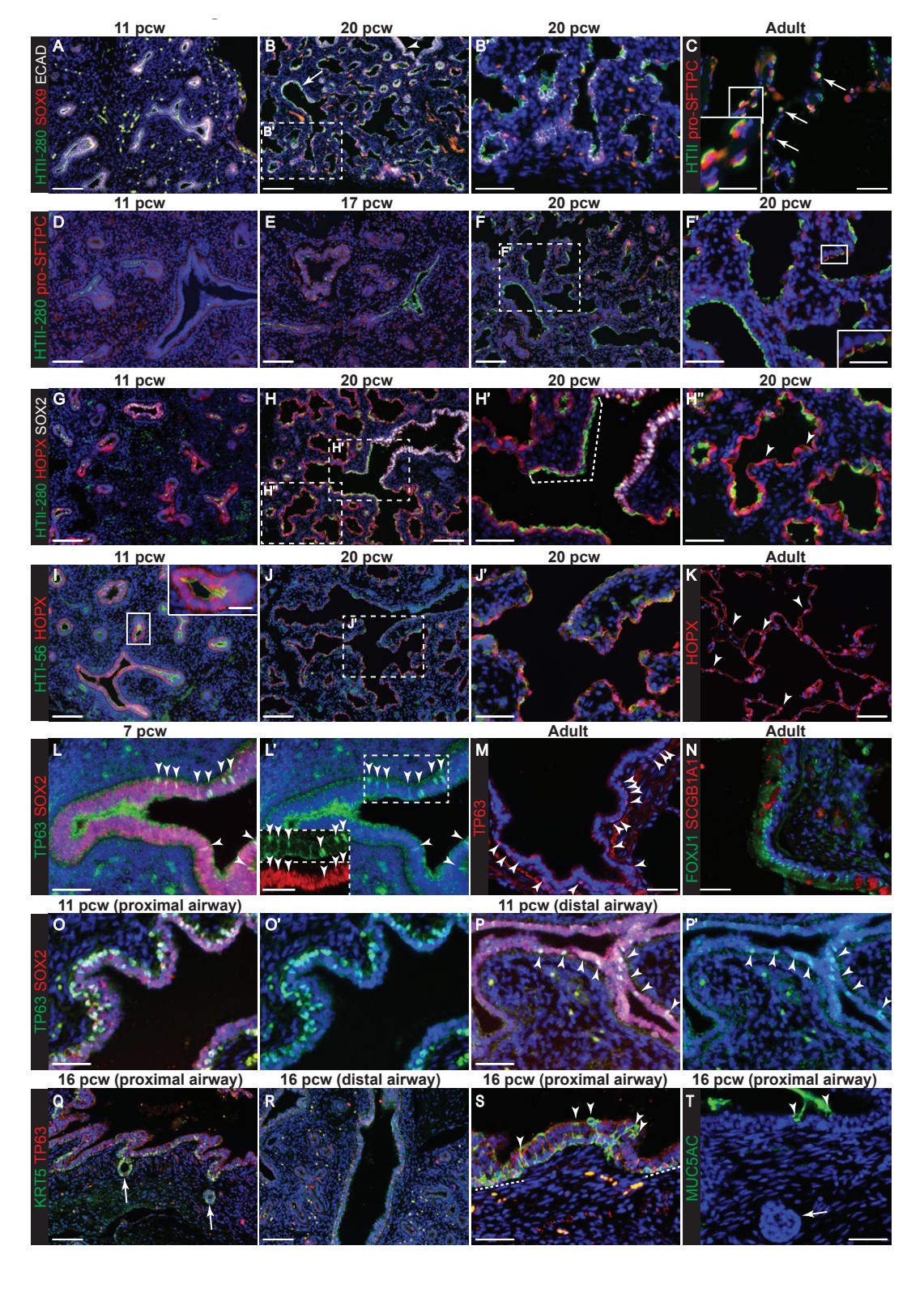

**Figure 1.** Evolution of alveolar and bronchiolar marker gene expression during human embryonic lung development. Sections of human embryonic and adult lungs. (**A, B**) 11 and 20 pcw. Green: HTII-280; red: SOX9 (tips); white: ECAD (epithelial cells). Arrow = HTII-280 positive terminal airway. (**C**) Adult. Green: HTII-280 (AT2 cells); red: pro-SFTPC (AT2 cells). Arrow = selected AT2 cells. (**D–F**) 11, 17 and 20 pcw. Green: HTII-280; red: pro-SFTPC. (**G–H**) 11 and 20 pcw. Green: HTII-280; red: HOPX; white: SOX2 (bronchiolar cells). Dotted bracket = terminal airway cells co-expressing SOX2 and HTII-280.

*Figure 1 continued on next page*

*Figure 1 continued*

Arrowheads = differentiating AT1 cells. (**I, J**) 11 and 20 pcw. Green: HTI-56; red: HOPX. **K.** Adult. Red: HOPX (AT1 cells). Arrowheads = HOPX⁺ nuclei. (**L**) 7 pcw. Green: TP63; red: SOX2. Arrowheads = TP63/SOX2 dual-positive cells in the more proximal airway region. Boxed region is shown as an inset in L' with channels separated. (**M**) Adult. Red: TP63 (basal cells). Arrowheads = TP63⁺ cells in an intra-lobar bronchiole. (**N**) Adult. Green: FOXJ1 (ciliated cells); red: SCGB1A1 (secretory cells). (**O, P**) 11 pcw proximal and distal airway from the same lung. Green: TP63; red: SOX2. Arrowheads = TP63/SOX2 dual-positive cells in the more distal airway. (**Q–S**) 16 pcw proximal and distal airway from the same individual. Green: KRT5; red: TP63. Arrows = developing sub-mucosal glands. Arrowheads = KRT5⁺, TP63⁻ cells. Dashed lines = patches of KRT5⁺, TP63⁺ cells. (**T**) 16 pcw proximal airway. Green: MUC5AC. Arrows = developing sub-mucosal gland. Arrowheads = mucous cells. Blue: DAPI (nuclei). Bars = 100 μm (**A,B,D,E,F,G,H,I,J,Q.R**); 50 μm (**B', C, F',H',H'',J',K,L,L',M,N,O,O',P,P',S,T**); 25 μm (insets in **C,F',I,L'**).

The following figure supplements are available for figure 1:

**Figure supplement 1.** Representative morphology of human embryonic lung samples.

**Figure supplement 2.** The evolution of alveolar marker gene expression in the human embryonic lung.

**Figure supplement 3.** Epithelial PDPN expression is not specific to AT1 cells during human lung development.

**Figure supplement 4.** Epithelial AQP5 expression is not specific to AT1 cells during human lung development.

**Figure supplement 5.** NKX2-1 and FOXA2 are expressed in all human lung epithelial cells up to 20 pcw.

(*Figure 1—figure supplement 2C*). In adult human alveoli a relatively small number of HOPX⁺ nuclei can be observed whilst the entire alveolar surface is covered by HOPX⁺ cell membranes extending from the AT1 cells (*Figure 1K*). PDPN (Podoplanin, also known as T1α) protein is detected in mouse lung epithelium from the time of alveolar specification at ~E16.5 and becomes restricted to differentiating AT1 cells by E18.5 (*Laresgoiti et al., 2016*). It is also a marker of airway basal cells and lymphatic endothelium (*Breiteneder-Geleff et al., 1999*; *Farr et al., 1992*). We detected low levels of apical PDPN expression in columnar (not basal) cells of the developing human bronchioles from 9 pcw (*Figure 1—figure supplement 3A*). At later stages expression is stronger in the more distal regions where PDPN and HTII-280 are co-expressed ubiquitously on the developing alveolar surface at 17 pcw, but PDPN is absent from the distal tips (*Figure 1—figure supplement 3B–G*). By 20 pcw PDPN can be observed to be co-expressed with AQP5 (Aquaporin 5) in distinct, individual cells, rather than running continuously through the alveolar duct (*Figure 1—figure supplement 3H*). It is strongly expressed in adult AT1 cells as expected (*Figure 1—figure supplement 3H*). AQP5 has been described in mouse as specifically detected in differentiating AT1 cells and not expressed in the bipotent alveolar progenitors which co-express AT1 and 2 markers (*Desai et al., 2014*). It is therefore a putative marker of differentiating AT1 cells. We first detect AQP5 in distal regions of the lung from 11 pcw (*Figure 1—figure supplement 4A–C*). By 20 pcw it can be observed to be expressed ubiquitously in the low columnar epithelium that lines the developing alveolar ducts and extends to the distal tips. It is expressed at higher levels in elongating cells with very short membrane extensions (*Figure 1—figure supplement 4D,E*; these are similar in appearance to the HOPX⁺ cells marked with arrowheads in *Figure 1H''* and are likely to be differentiating AT1 cells that are beginning to extend their membranes). In the adult alveoli AQP5 is highly expressed in AT1 cells, but it can also be observed on the surface of non-ciliated cells in the lower airways (*Figure 1—figure supplement 4F,G*).

NKX2-1 is ubiquitously expressed in lung epithelial cells from the time of lung specification in the foregut (*Lazzaro et al., 1991*). NKX2-1 levels have been reported to be heterogeneous in mature alveolar cells in the adult mouse and rat (*Desai et al., 2014*; *Liebler et al., 2016*). We observed that NKX2-1 was co-expressed with FOXA2 in all human embryonic lung epithelial cells from 9 to 21 pcw, albeit at higher levels in alveolar-fated than bronchiolar-fated cells (*Figure 1—figure supplement 5A–C*). NKX2-1⁻,FOXA2⁻ cells which apparently lined the more mature alveolar ducts were identified, but closer inspection revealed that these were ECAD⁻ non-epithelial cells their presence at the alveolar surface likely being an artefact due to the angle of sectioning (*Figure 1—figure supplement 5D–G*).

Our results show that the broad picture of mouse alveolar development (low-level co-expression of AT1 and 2 markers in cells that exit the canalicular stage distal tip, followed by lineage-specific expression as morphogenesis proceeds) is conserved in human. However, even in this limited analysis there are striking mouse-human differences. The timing of alveolar marker gene onset in human is ~11 pcw, preceding the canalicular stage by approximately five weeks. In addition, the relative timing of expression of specific markers can differ (pro-SFTPC, LPCAT1, AQP5). Moreover, the developing human terminal bronchioles are SOX2$^+$, but co-express alveolar markers which to our knowledge has never been observed in mouse (*Laresgoiti et al., 2016*).

The first signs of cellular heterogeneity/differentiation in the bronchioles were observed from 7 pcw where we saw that a sub-set of SOX2$^+$ cells co-expressed low levels of TP63 (Tumour Protein P63) in the larger airways (*Figure 1L*, arrowheads). These are likely to be differentiating basal cells which are TP63$^+$ in the adult (*Figure 1M*, arrowheads) (*Rock et al., 2009*). FOXJ1 and SCGB1A1 were also readily detected in adult airways (*Figure 1N*). By 11 pcw, TP63 staining was much stronger and localised to a sub-set of basally-located airway nuclei (*Figure 1O*). TP63 was also stronger in more proximal, versus distal, bronchioles within the same lungs (*Figure 1O,P*). We first detected KRT5 (Keratin 5) at 16 pcw in the larger airways, particularly in invaginating submucosal gland buds (*Figure 1Q–S*). KRT5 was also seen in patches of TP63$^+$ airway cells, likely differentiating basal cells (*Figure 1—figure supplement 1S*, dotted lines). KRT5$^+$, TP63$^-$ cells were also observed, frequently in a non-basal position (*Figure 1—figure supplement 1S*, arrowheads). The first signs of columnar cell differentiation were detected at 16 pcw when rare MUC5AC$^+$ (Mucin 5AC) cells were identified in the proximal airways (*Figure 1T*). This is consistent with a proximal-distal pattern of airway differentiation as in other species (*Plopper et al., 1992*; *Toskala et al., 2005*).

## Characterization of human embryonic lung tip progenitors

In the pseudoglandular stage mouse lung, there is a clear demarcation between SOX9$^+$ tip and SOX2$^+$ stalk (differentiating bronchiole) cells and multiple signalling mechanisms regulate the boundary between the two populations (*Hrycaj et al., 2015*; *Mahoney et al., 2014*; *Wang et al., 2013*). We observed a tip-stalk boundary in the pseudoglandular stage human lungs with SOX9 restricted to the tip and SOX2 expressed highly in the stalk. Differentiating αSMA$^+$ (α-Smooth Muscle Actin) smooth muscle was observed around the SOX2$^+$ future airways and SOX9 was present throughout the mesenchyme at low levels (*Figure 2A*). We noted that human tip epithelium was more proliferative than stalk (*Figure 2B*), consistent with mouse results (*Okubo et al., 2005*). However, in contrast to the mouse, low levels of SOX2 were co-expressed with SOX9 in the distal tip epithelium throughout the pseudoglandular stage (*Figure 2B*; *Figure 2—figure supplement 1A–C*). *SOX2*, *SOX9* co-expression at the tip was confirmed by qRT-PCR in microdissected tip and stalk cells (*Figure 2—figure supplement 1D*). Further examination of our time-course revealed that SOX2 gradually decreased over time and disappeared from the tip epithelium at the transition to the canalicular stage of development. This happened heterogeneously throughout the lung. For example, at 17 pcw we observed a mixture of SOX2$^+$ and SOX2$^-$ distal tips within individual lungs (*Figure 2C,D*; *Figure 2—figure Supplement 2*). However, by 20 pcw all distal tips were SOX2$^-$ (*Figure 2E*; *Figure 2—figure Supplement 2*). Moreover, there was a SOX2$^-$, SOX9$^-$ zone adjacent to the 20 pcw distal tips which corresponds to the developing saccules where markers of alveolar differentiation are expressed (compare *Figure 2E* with *Figure 1H*).

To compare tip-stalk gene expression globally we microdissected tip and stalk epithelium from four individual 6–7 pcw lungs and performed RNAseq (*Figure 2F*). We detected a total of 15,599 transcripts, with the majority (86%) expressed in both tip and stalk (*Figure 2G*). This high level of similarity was expected since the stalk cells are the immediate descendants of the tip population. Using a two-fold difference in expression level cut-off, 2208 genes that were enriched in stalk or tip were also identified. Non-hierarchical clustering of the samples based on these 2208 transcripts revealed a clear separation into distinct tip and stalk populations (*Figure 2H*). Moreover, these genes were expressed at similar levels in published whole human foetal lung RNAseq (*Figure 2H*) (*Bernstein et al., 2010*). Gene Ontology (GO) analysis of the differentially expressed transcripts included categories related to cell signalling, proliferation, adhesion, motility, transcription and developmental processes expected for embryonic progenitors (*Figure 2I*). These GO categories were also a major feature of genes that were co-expressed in both tip and stalk (*Figure 2—figure*

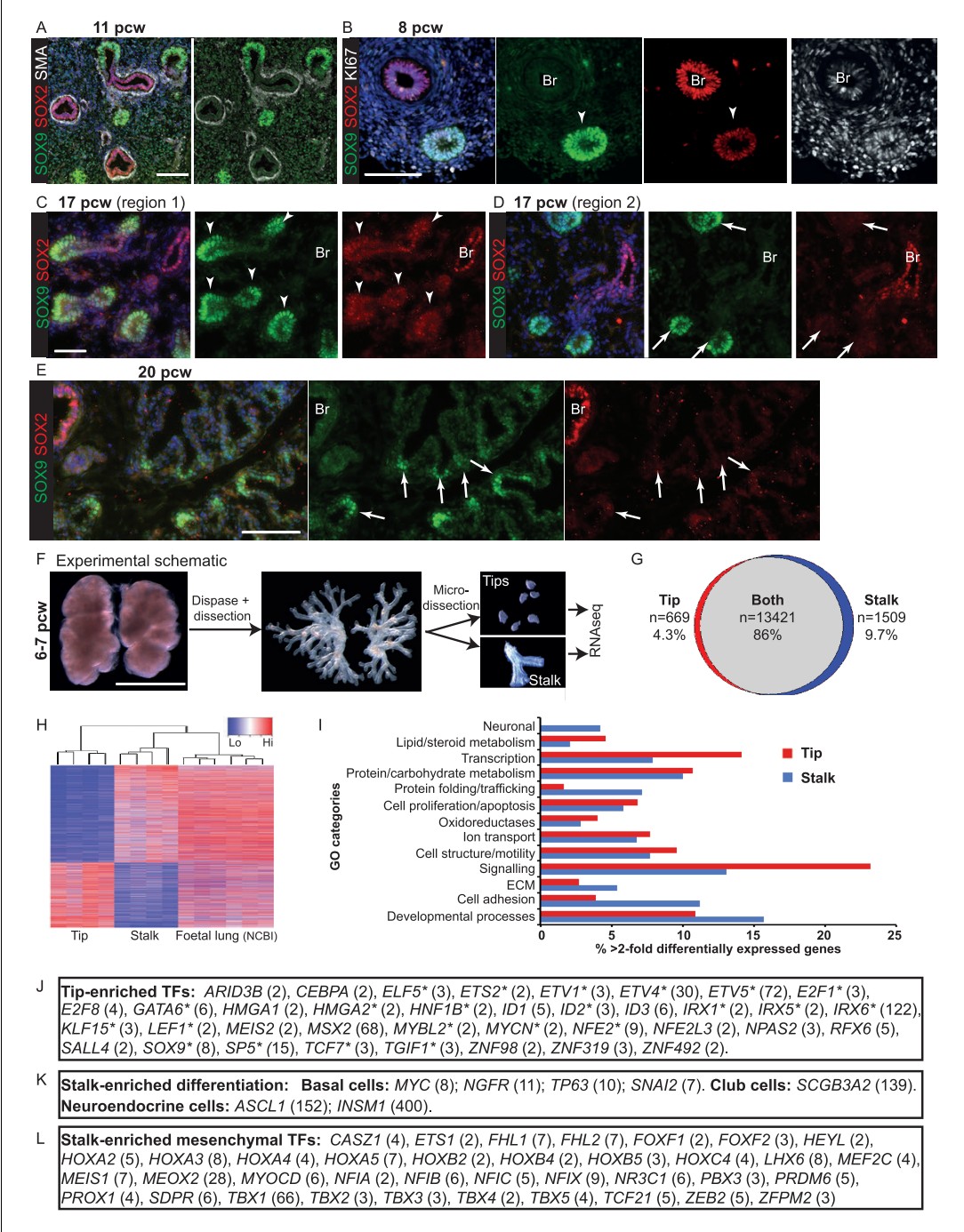

**Figure 2.** The tip and stalk epithelial cell populations are clearly demarcated in branching human, pseudoglandular stage, lungs. (A–E) Sections of human embryonic lungs. (A) 11 pcw. Green: SOX9 (tip); red: SOX2 (stalk); white: α-SMA (smooth muscle). (B) 8 pcw. Green: SOX9 (tip); red: SOX2 (stalk); white: KI67 (proliferating cells). (C, D) 17 pcw. (E) 20 pcw. Green: SOX9 (tip); red: SOX2 (stalk). Arrowheads = SOX9+, SOX2- co-expressing tips. Arrows = SOX9+, SOX2- tips. Blue: DAPI (nuclei). (F) Experimental schematic for tip versus stalk RNAseq. (G) Venn diagram showing common and differentially-expressed transcripts based on a fold-change of at least 2. (H) Unsupervised hierarchical clustering of tip, stalk and published foetal lungs based on the differentially-expressed genes. (I) Chart to show the percentage of the gene ontology classes represented in the differential expression data. (J) List of transcription factors enriched at least two-fold in the tips. Number in brackets indicates fold-change over the stalk. * indicates reported mouse tip expression, see *Supplementary file 2*. (K) List of differentiation markers enriched in the stalk. (L) List of transcription factors enriched in stalks that were previously reported as expressed in the mesenchyme . Scale bars = 50 μm (A, C, D); 100 μm (B, E); 2 mm (F).

The following figure supplements are available for figure 2:

*Figure 2 continued on next page*

*Figure 2 continued*

**Figure supplement 1.** Pseudoglandular stage human lung tips co-express SOX9 and SOX2.

**Figure supplement 2.** Human lung tips down-regulate SOX2 during the canalicular stage.

**Figure supplement 3.** Transcriptional differences and similarities between human pseudoglandular stage tip and stalk populations.

**Figure supplement 4.** Specific transcription factors are enriched at the protein level in human distal epithelial lung tips.

**Figure supplement 5.** Tip gene expression is highly conserved between mouse and human.

*supplement 3A*). *NKX2.1* and *FOXA2,* which we detect as ubiquitous throughout the developing human lung epithelium (*Figure 2—figure supplement 3B*), were not enriched in tip or stalk.

The human tip-enriched data set contained 37 genes annotated as transcription factors of which 54% (20/37) had previously been characterized as expressed in mouse tips (*Figure 2J*; *Supplementary files 1* and *2*). These included *GATA6, HMGA2, MYCN* and *SOX9* which have documented tip-specific functions in mouse (*Chang et al., 2013*; *Rockich et al., 2013*; *Singh et al., 2014*; *Zhang et al., 2008*). Moreover, we were able to confirm human tip-enrichment of ETV5, HMGA1, HMGA2, HNF1B and ID2 at the protein level (*Figure 2—figure supplement 4*). Among the tip transcription factors, we identified only one gene, *MEIS2*, which is likely to be mesenchymally expressed (*Diez-Roux et al., 2011*; *Herriges et al., 2012*), suggesting a very low level of mesenchymal contamination. A comparison of our human tip RNAseq with a previously published mouse tip microarray (*Laresgoiti et al., 2016*) found that 96% of orthologous genes that were expressed in human tips were also present in mouse (*Figure 2—figure supplement 5A*). There was also a good correlation between levels of mouse-human orthologous gene expression (*Figure 2—figure supplement 5B*).

We detect the first signs of cellular heterogeneity in the stalk by 7 pcw (*Figure 1L*). Consequently, we were able to identify stalk-specific transcripts that are characterized as expressed in differentiating airway cells (*Figure 2K*). These included, basal, neuroendocrine and club cell markers (*Guha et al., 2012*; *Ito et al., 2000*; *Jia et al., 2015*; *Lan and Breslin, 2009*; *Rock et al., 2009*). Consistent with this, we observed neuroendocrine cells in pseudoglandular stage airways (*Figure 2—figure supplement 3C,D*). When we examined the stalk transcripts annotated as transcription factors in more detail, we noticed that 52% (67/128) had published mouse airway expression patterns. However, 55% of these (37/67) were expressed in mouse mesenchyme, rather than epithelium (*Figure 2L*), suggesting a high level of mesenchymal contamination in the stalk samples.

Within the RNAseq data, we identified components of the EGF, FGF, Hedgehog, IGF, Notch, Retinoic Acid, TGF-$\beta$ super-family and WNT signalling pathways (*Supplementary file 1*). We noted that while core downstream signalling components were transcribed in both tip and stalk; ligands, receptors and inhibitors were more likely to be tip or stalk specific. FGF Receptor signalling is central to mouse lung branching morphogenesis (*Volckaert and De Langhe, 2015*) and protein expression of several FGF pathway components was confirmed by antibody staining (*Figure 2—figure supplement 3E–G*). There were various subtle differences in tip signalling pathways between mouse and human. For example, *BMP2* and *BMP7* were highly enriched in human tips compared with *Bmp4* in the mouse (*Bellusci et al., 1996*) and *IHH* in human where mouse has *Shh* (*Bellusci et al., 1997*).

This analysis suggests that the human tip epithelium is analogous to the mouse population with a highly conserved transcriptome and similar signalling pathway activity. However, we also observe differences that are likely to be functionally significant.

## Establishment of in vitro self-renewing culture conditions for human epithelial tips

The mouse distal tip population is a long-lived progenitor that self-renews extensively throughout normal lung development. Moreover, our recent heterochronic grafting experiments demonstrated that its behaviour is largely controlled by extrinsic signals (*Laresgoiti et al., 2016*). We therefore reasoned that we should be able to capture tip self-renewing behaviour in vitro by supplying the correct

combination of factors. This would be analogous to the long-term self-renewal of blastocyst inner cell mass as ES (Embryonic Stem) cells. We microdissected human epithelial tips from 5 to 9 pcw lungs (as in *Figure 2F*) and plated them in Matrigel in the presence of 7 factors: EGF, FGF7, FGF10, NOG (Noggin), RSPO1 (R-spondin 1), a GSK3$\beta$ inhibitor CHIR99021 and a TGF$\beta$ inhibitor SB431542. Factor choice was based on the conditions used to grow adult foregut derivatives as organoids, the extensive literature on mouse lung development and our RNAseq analysis (*Huch et al., 2013a*, *2013b*; *Sato et al., 2009*; *Swarr and Morrisey, 2015*; *Yin et al., 2014*). In these conditions human lung epithelial tips formed organoids with 100% colony forming efficiency (n = 303 tips from 13 individuals). Tips formed spheres within 12 hr, expanded spherically for 6–8 days and then branched; by culture day 14 the organoids resembled a mass of tips (*Figure 3A*; *Video 1*). We passaged the organoids every 2 weeks by mechanically breaking into smaller pieces and re-plating. Growth continued in a similar fashion in later passages, although the morphological appearance of the cultures became more heterogeneous depending on the extent of breakage during passaging (*Figure 3A*). Tip organoids retained SOX2 and SOX9 expression over multiple passages (*Figure 3B*). Moreover, they expressed the lung-specific transcription factor NKX2-1 and the tip-specific marker proteins that we have validated (*Figure 3—figure supplement 1*). Organoids possessed a lumen and were composed of a single epithelial layer (*Video 2*) similar to the in vivo morphology of the tip epithelium (*Figure 2—figure supplement 1B,C*). Every tip organoid line we isolated continued to grow in the same way for at least 9 passages over 4 months (n = 11 organoid lines were maintained for >9 passages without apparent change in morphology, or SOX2 and SOX9 expression; some cultures have been maintained for up to 9 months). Organoid karyotypes were also normal (*Figure 3—figure supplement 2*). We concluded that activation of EGF, FGF and WNT signalling, and inhibition of BMP and TGF$\beta$, are sufficient to grow human epithelial tips as long-term, self-renewing organoids with an initial colony forming efficiency of 100%.

We have recently shown that differentiating mouse bronchiolar cells can produce alveolar descendants (*Laresgoiti et al., 2016*). We therefore tested if human stalk cells, which are undergoing the first signs of bronchiolar differentiation (*Figure 1L*; 2K), could be induced to grow as tip-like organoids using our tip culture conditions. The human stalks formed organoids with a 100% colony forming efficiency (n = 17 stalks from five individuals). These grew spherically, branched with a similar morphology to tip organoids and could be passaged over many months (*Figure 3C*). Consistent with their branching behaviour, by the end of P0 the SOX2$^+$ stalks had become SOX2$^+$, SOX9$^+$ organoids and they retained expression of both proteins over multiple passages (*Figure 3D*). Similarly, we asked if the SOX9$^+$, SOX2$^-$ tip cells from 19 pcw lungs (*Figure 2E*) could be cultured as self-renewing organoids in the same conditions. The 19 pcw tips also grew as branching organoids for multiple passages with a 100% colony forming efficiency (organoids from three individuals maintained for at least six passages). The 19 pcw tips also became SOX2$^+$, SOX9$^+$ organoids in vitro (*Figure 3—figure supplement 3*). Thus the tip organoid self-renewal conditions are sufficient to convert SOX2$^+$, SOX9$^-$ stalks and SOX2$^-$, SOX9$^+$ tips into long-term self-renewing SOX2$^+$, SOX9$^+$ organoids.

As a first test of our in vitro self-renewing organoid system, we compared the transcriptome of organoids to that of freshly-isolated tips and stalks. When the whole transcriptome is compared freshly isolated tips and stalks can be separated into two distinct clusters on a multi-dimensional scaling plot (*Figure 3E*), although it should be remembered that they are very similar cell types (*Figure 2G*). The organoid transcriptomes lie between the freshly isolated tips and stalks, showing that they retain many transcriptional features of the cells from which they are derived (*Figure 3E*). Analysis of markers that we have characterized as tip or stalk enriched supports this conclusion (*Figure 3F*, *Figure 3—figure supplement 4*). Thus the organoids are transcriptionally very similar to the starting progenitor population, with a tendency to also express stalk markers. We could detect no signs of organoid differentiation into mature lung cell lineages, or towards non-lung cell types.

We next asked if our human tip organoid growth conditions were sufficient to support the growth of mouse tips as long-term self-renewing organoids. We plated E12.5 mouse tips in identical growth conditions, and also without SB431542. In both media conditions mouse tips formed branching organoids with 100% efficiency (*Figure 3—figure supplement 5*). Mouse organoids grew at a faster rate than the human organoids and were initially SOX9$^+$, SOX2$^-$, reflecting the in vivo mouse situation. However, by passage six the mouse tip organoids had decreased SOX9 and started to express SOX2 (*Figure 3—figure supplement 5B,D*). We concluded that the human growth conditions are not sufficient to support the long-term self-renewal of undifferentiated mouse tip cells, consistent

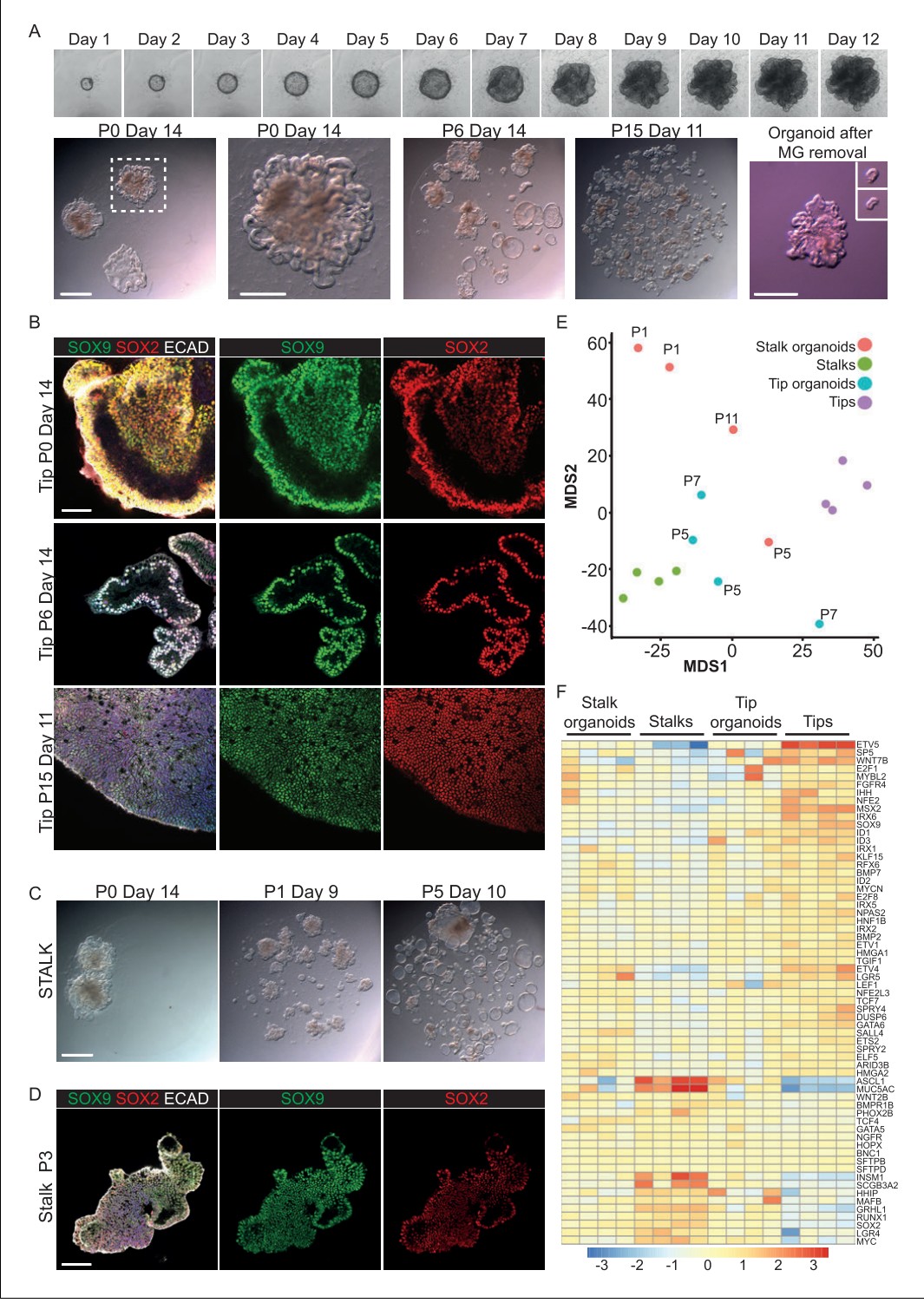

**Figure 3.** Long-term, self-renewing organoid culture of human lung epithelial tip cells with a initial colony forming efficiency of 100%. (**A**) Frames from *Video 1* showing bright field images of a single microdissected tip taken every 24 hr for 12 days. Representative bright field images of tip organoid cultures from P0, P6 and P15. A typical organoid after matrigel removal is shown and after further microdissection of branched structures (inset). (**B**) Confocal images of tip organoids at P0, P6 and P15. Green: SOX9; red: SOX2; white: ECAD. (**C**) Bright field images of stalk organoids cultured in self-renewing medium at P0, P1, P5. (**D**) Confocal images of stalk organoids at P3. Green: SOX9; red: SOX2; white: ECAD. (**E**) Multidimensional scaling plot showing the distribution of fresh

*Figure 3 continued on next page*

*Figure 3 continued*

tip and stalk transcriptomes and cultured organoids. (F) Heat map of selected tip, stalk and differentiation markers. Bars = 1 mm (A, C); 50 μm (B, D).

The following figure supplements are available for figure 3:

**Figure supplement 1.** Organoids passaged in self-renewing medium retain tip-specific transcription factor proteins.

**Figure supplement 2.** Organoids passaged in self-renewing medium retain a normal karyotype.

**Figure supplement 3.** SOX2⁻, SOX9⁺ canalicular stage human embryonic tips can be grown as SOX2⁺, SOX9⁺ self-renewing organoids.

**Figure supplement 4.** Box plots of selected tip and stalk specific genes showing transcript levels in fresh tissue and cultured organoids.

**Figure supplement 5.** E12.5 mouse tips do not long-term self-renew in the growth medium developed for human tips.

with our data that there are differences in signalling gene expression between mouse and human tip epithelium.

## Efficient organoid establishment required all seven factors contained within the growth medium

During organoid establishment diffuse mesenchymal cells were always visible in the cultures and more prevalent in the stalk-derived organoids. We estimated the proportion of mesenchymal cells in freshly microdissected tips and stalks (*Figure 4A,B*). There was $0.04 \pm 0.07\%$ mesenchyme in the tip dissections and $3.8 \pm 0.47\%$ in the stalk (n = 4 samples; mean ± standard deviation). This estimate is in agreement with the observation that mesenchymal genes were present in the RNAseq of micro-dissected stalk (*Figure 2J,L*). We therefore cannot exclude that the presence of the mesenchyme plays a role in the establishment of organoid cultures. However, when we looked for mesenchyme in

our passaged human tip organoids, we were able to observe ECAD⁻ cells at P0, P1 and, rarely, at P2, but never in higher passage number organoids (*Figure 4C*). Thus, mesenchymal cells are not required for organoid maintenance.

We focussed on human tip epithelium to determine whether all seven factors (EGF, FGF7, FGF10, NOG, RSPO1, CHIR99021 and SB431542) were required for organoid establishment. We plated tips from the same lung in our self-renewing conditions, or in eight other media combinations in which specific factors were removed. To our surprise, SOX2⁺,SOX9⁺ organoids grew in all conditions tested (*Figure 4D*; n = 4 experiments with different individual lungs). However, compared to our self-renewing condition, all other media produced smaller organoids. These organoids could mostly be passaged (except in the absence of CHIR99021 where organoids disintegrated at passaging), but they remained extremely small at P1 and subsequent passaging was not attempted (*Figure 4E*). Interestingly, in the absence of

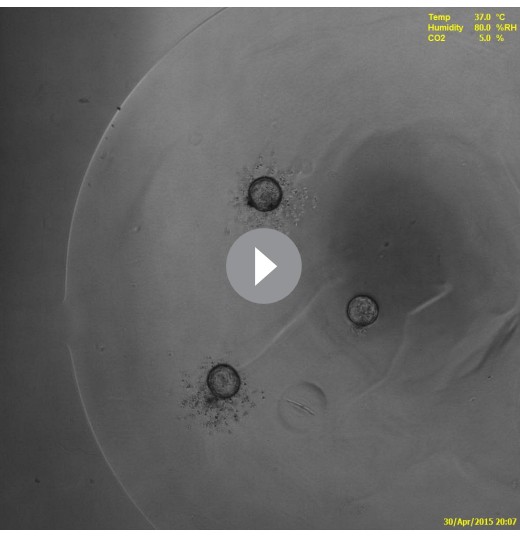

**Video 1.** Human lung epithelial tip growing into an organoid over 11 days. Imaged every 12 hr in bright-field on a Nikon Biostation.

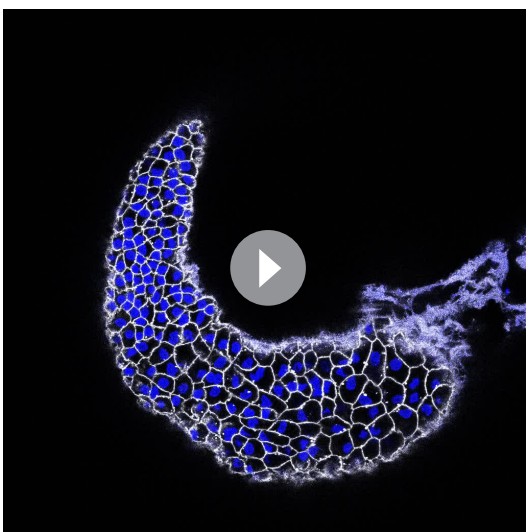

**Video 2.** Organoid structure is a single layer epithelium with a hollow lumen. Confocal z-stack of P6 tip organoid. Nuclei (DAPI, blue) and epithelial cells (ECAD, white) illustrating typical organoid morphology.

NOG the mesenchymal cells present in the cultures expanded much more than in self-renewing medium. These mesenchymal cells were initially SOX9$^+$, as in vivo, but lost SOX9 expression by P1. Whereas, in the absence of FGF10 we noted that by P1 the organoids had distinct SOX2$^+$, SOX9$^+$ and SOX2$^-$, SOX9$^+$ domains (*Figure 4E*), suggesting that regionalization was occurring spontaneously. We repeated organoid derivation in self-renewal medium, versus self-renewal medium without SB431542, and quantified the effects on organoid-forming efficiency and size. In the presence of SB431542 organoid-forming efficiency was 100% and organoids were larger (*Figure 4F*). It has recently been noted that growth of human organoids usually requires TGF$\beta$ inhibition, whereas mouse organoids usually do not (*Huch and Koo, 2015*).

## FGFR and WNT signalling are required to maintain *SOX9* during organoid self-renewal

We grew established organoid lines for three days in media in which one or more of the factors was altered to test the effects on *SOX2* and *SOX9* levels by qRT-PCR (*Figure 4G*; n = 3 experiments on three different tip organoid lines). None of the conditions resulted in any effect on organoid morphology over the 3 days. Growth in basal medium caused a significant reduction in *SOX9* and a less-reproducible increase in *SOX2*; consistent with a major function of the factor combination to promote tip, and inhibit bronchiolar, fate. Removal of SB431542, or FGF7 and FGF10, or CHIR99021 caused a significant reduction in *SOX9* levels. Moreover, removal of FGF7 and FGF10, or CHIR99021, increased *SOX2*. Removal of FGF7, or 10, alone had no significant effects. These data suggest that FGF and WNT signalling are both required to promote tip self-renewal at the expense of differentiation, consistent with known functions in mouse (*Volckaert et al., 2013*).

## Differentiation of human tip organoids in the presence of mouse embryonic or adult lung cells

To functionally test if the tip organoids retained their lung identity after long-term culture, we tested if they were capable of integrating into adult mouse lungs in vivo. We injured immune-compromised *NOD-scid-IL2rg$^{-/-}$* (NSG) mice with a low dose of bleomycin and intra-tracheally transplanted 6 × 10$^5$ single cells isolated from self-renewing organoids (*Figure 5—figure supplement 1A,B*). At day two post-transplant small groups of cells were visible in 100% (4/4) of transplanted mice and these had grown into much larger patches by day 8 (*Figure 5—figure supplement 1C–E*). Grafts were usually observed in the region spanning the bronchioles and alveoli. Human cells retained SOX2 and SOX9 co-expression (*Figure 5—figure supplement 1G*). However, they frequently turned off NKX2-1 but retained FOXA2 (*Figure 5—figure supplement 1F*). They showed the first signs of airway differentiation with sub-sets of graft cells expressing KRT5, TRP63 and MUC5AC (*Figure 5—figure supplement 1H,I*). These data strongly suggest that lung identity is retained within the organoids. However, it is not known if embryonic human cells engrafted into adult mouse lungs can receive appropriate differentiation cues. We therefore performed a similar set of experiments using the kidney capsule environment for chimeric human-mouse embryonic grafts.

E13.5 mouse lungs were dissociated, mixed with dissociated human tip organoids, formed into a cell pellet and transplanted under the kidney capsule of NSG mice (*Figure 5A*). All kidneys harvested contained mouse/human chimeric lung grafts (9/9) in which the mouse and human cells tended to segregate (*Figure 5B*). The mouse parts of every graft contained either squamous

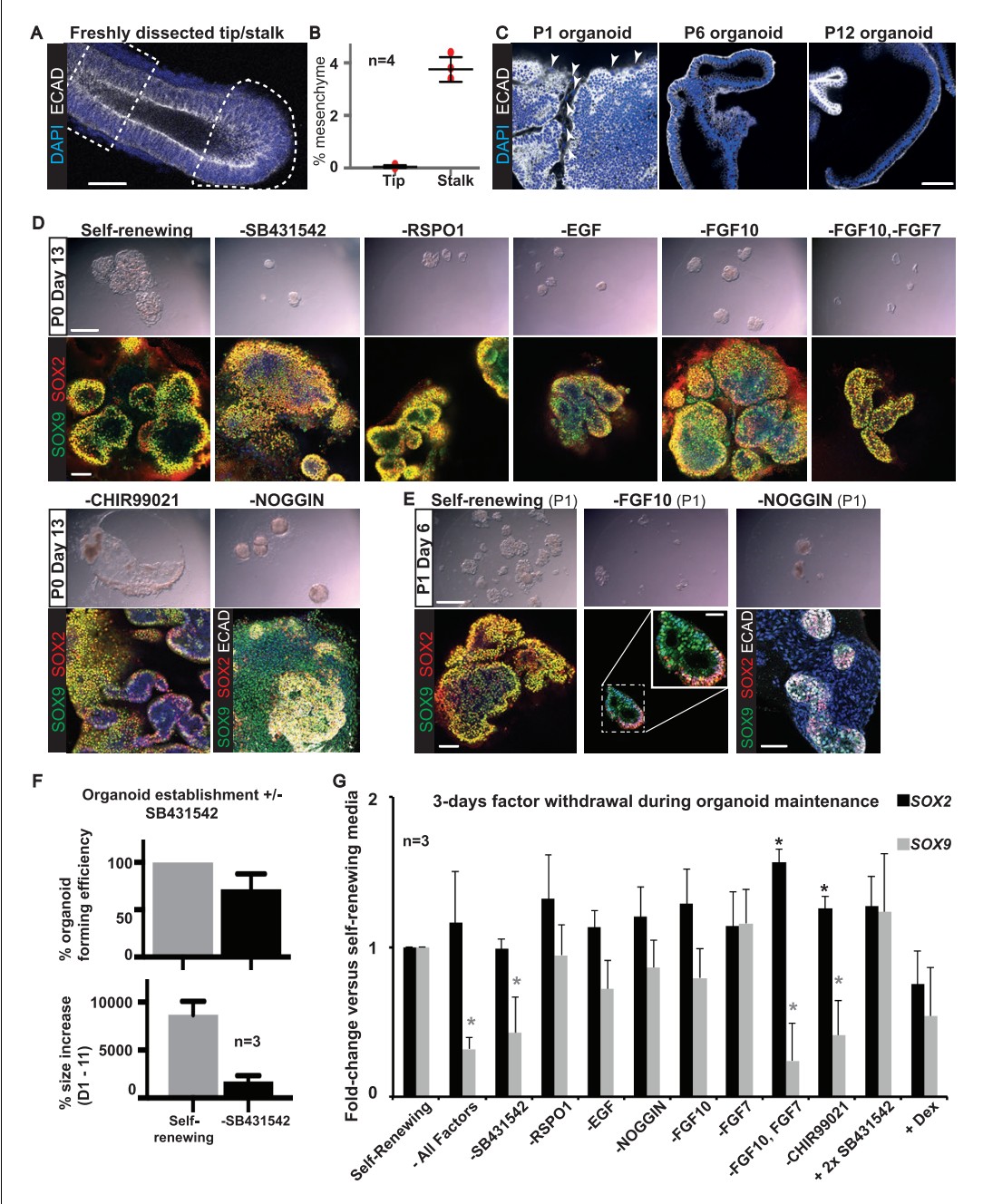

**Figure 4.** All factors added to the medium are required for culture establishment. (A) Whole-mount staining for DAPI and ECAD was performed to estimate the fraction of mesenchyme in a microdissected tip (dotted area) and stalk (dashed box). (B) Quantitation of percentage of mesenchyme in four microdissected tips and stalks. (C) Presence of mesenchyme was assessed in six organoid lines over multiple passages by staining for ECAD (arrowhead = mesenchymal cells). (D) Bright field and confocal images of P0 Day 13 organoids cultured in self-renewing medium, or without the indicated factors. Green: SOX9; red: SOX2; white: ECAD. (E). Bright field and confocal images of organoids at P1 Day 6 cultured in self-renewing medium, or without FGF10, or Noggin. Green: SOX9; red: SOX2; white: ECAD. Boxed area is magnified in inset. (F) Quantitation of organoid forming efficiency and size with, or without, TGFβ inhibition. Bars = SEM. Three biological replicates were analysed; 61 tip cultures without TGFβ inhibition, 38 tip cultures with TGFβ inhibition. (G) Established organoid lines were grown for 3 days in self-renewing, or indicated test medium, and levels of *SOX2* and *SOX9* assessed by qRT-PCR (values normalized to one for self-renewing controls). Three independent tip organoid lines at P9, 15 and 21 were used. Bars = SEM. * = p-value<0.05. Bars = 50 μm (A, C; D, E confocal images); 1 mm (D, E bright field).

The following source data is available for figure 4:

**Source data 1.** Individual data points for *Figure 4B*.

*Figure 4 continued on next page*

*Figure 4 continued*

**Source data 2.** Individual data points for *Figure 4F*.
**Source data 3.** Raw qRT-PCR data for *Figure 4G*.

epithelium with differentiated alveolar cells (*Figure 5B*; Supplement 2A,B), or columnar epithelium with differentiated bronchiolar cells (*Figure 5—figure supplement 2C–E*). By contrast, human cells were always assembled into columnar epithelial airway-like structures surrounded by mouse mesenchyme. Rare human pro-SFTPC$^+$ cells were found in all samples (*Figure 5C*; *Figure 5—figure supplement 2F*). Human cells were clearly SOX9$^+$, SOX2$^+$ at 3 weeks, although patches of cells downregulating SOX9 were visible (*Figure 5D*). By 7 and 12 weeks SOX9 was expressed at very low levels in only some human cells. Rare patches of differentiated human airway were identified from 2/3 organoid lines in kidneys harvested at 12 weeks. These were lined with basal, goblet and ciliated cells similar to the in vivo human airways (*Figure 5E,F*). The remaining human airway-like structures contained goblet cells only and were found in every graft.

These grafting experiments show that the organoids retain the ability to assemble into lung structures and robustly generate differentiated airway cells, but alveolar differentiation is either premature or ineffective. However, one limitation of these experiments is that the human cells may be unable to respond efficiently to the mouse differentiation signals; possibly because they require different signalling inputs. We therefore asked if it is possible to differentiate the human tip organoid lines to bronchiolar and alveolar fate in vitro.

## In vitro differentiation of tip organoids

To test the ability of our tip organoids to differentiate into bronchiolar structures we grew them for 2–4 weeks in an established human airway differentiation medium, PneumaCult. In high passage number organoids this mostly resulted in differentiation into SOX2$^+$, MUC5AC$^+$ goblet cells, although a smaller number of organoids also contained KRT5$^+$ basal cells (*Figure 6—figure supplement 1*; n = 3 organoid lines). In low passage number organoids, in which a small amount of mesenchyme was still present, growth in PneumaCult$^{TM}$ resulted in expansion of the mesenchyme and differentiation of TP63$^+$ basal cells, MUC5AC$^+$ goblet cells and rare ACT$^+$ ciliated cells in the epithelium (*Figure 6A–C*; n = 3, P2 organoid lines). SOX2, SOX9 co-expression was observed in the latter experiments, suggesting that airway differentiation was not complete.

There is no established culture medium for growing human alveolar epithelium. We therefore tested media conditions for their ability of turn off SOX2 and SOX9 and activate pro-SFTPC and/or HTII-280, compared to our self-renewing condition, over two weeks (*Figure 6—figure supplement 2*). SOX9 was lost in all conditions tested, possibly due to prolonged exposure to Dexamethasone which turns off *Sox9* in mouse lungs (*Figure 6—figure supplement 2C–E*) (*Alanis et al., 2014*). In general, we obtained somewhat patchy differentiation with regions where SOX2 was retained and regions where pro-SFTPC (or more rarely HTII-280) was expressed (*Figure 6—figure supplement 2F–N*). We next cultured organoids for 3 weeks in the most efficient alveolar medium (CHIR99021, FGF7, FGF10, Dexamethosone, cAMP, IBMX, T3, DAPT). This resulted in uniformly SOX2$^-$, SOX9$^-$, pro-SFTPC$^+$ organoids (*Figure 6E,F*; n = 2 organoid lines). These expressed low levels of pro-SFTPC and retained a columnar epithelial appearance, suggesting alveolar-fate rather than differentiation. HTII-280 expression was not detected.

We reasoned that addition of human canalicular stage mesenchyme would provide additional cues and promote improved alveolar differentiation. We isolated mesenchyme from 19 pcw human lungs by dispase digestion and micro-dissection, mixed it directly with human tip organoids and cultured in alveolar differentiation medium for three weeks. This resulted in heterogeneous organoids with patches of SOX2$^+$ cells and patches of pro-SFTPC$^+$ cells (*Figure 6G,H*; *Figure 6—figure supplement 3A–C*; n = 1 line for this preliminary experiment). Importantly, pro-SFTPC expression was stronger than without mesenchyme and the pro-SFTPC$^+$ cells had a more squamous appearance, similar to endogenous differentiating AT2 cells. However, there were many regions of the cultures where mesenchymal cells were not observed. To increase the proportion of mesenchyme in the co-

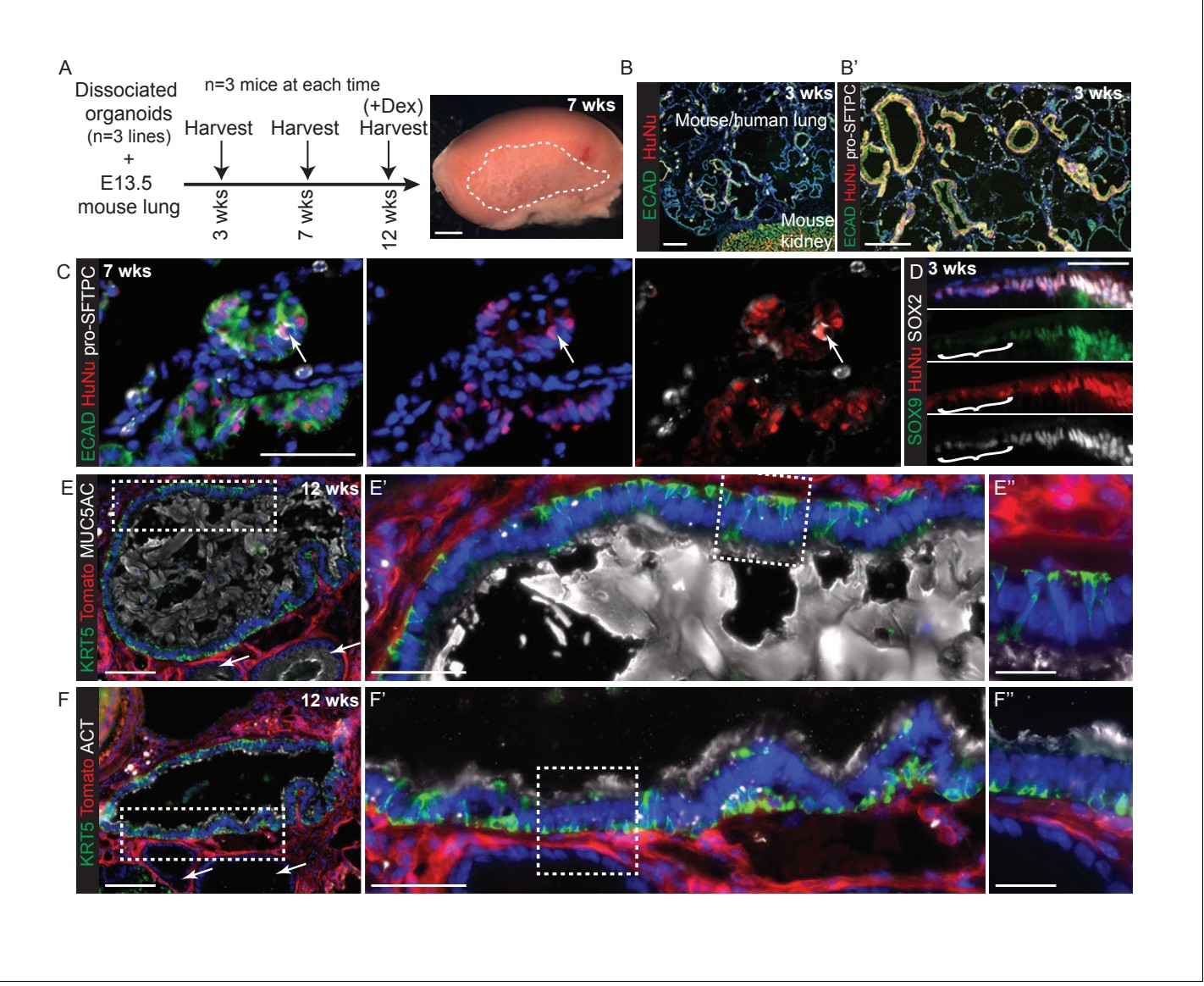

**Figure 5.** Disscoiated self-renewing organoids are competent to differentiate under the kidney capsule in the presence of embryonic mouse lung cells. (A) Experimental schematic. Dissociated organoids were mixed with dissociated E13.5 mouse lungs (either MF1 outbred strain for grafts harvested at 3 and 7 weeks, or *Rosa26R-mT-mG* strain for grafts harvested at 12 weeks.) Animals culled at 12 weeks received 3x dexamethasone injections 1 week before culling. The 7 and 12 week grafts looked identical and thus no effects on the grafts were observed from the dexamethasone injections. Three independent organoid lines were used and three mice (one per organoid line) culled at each time point. Grafts were clearly visible growing beneath the kidney capsule in all nine kidneys harvested. (B) Chimeric mouse/human lung structures were found in all kidneys. Green: ECAD (epithelium); red: HuNu (human nuclei). (B') It can clearly be seen that mouse and human cells tend to segregate within the grafts, possibly due to their differing size or surface properties. Wide-spread regions of mouse pro-SFTPC$^+$ cells were always visible. Green: ECAD; red: HuNu; white: pro-SFTPC. (C) Rare human pro-SFTPC$^+$ cells (arrow) were identified in all samples. Green: ECAD; red: HuNu; white: pro-SFTPC. (D) Human cells arranged into airway-like structures were strongly SOX9$^+$, SOX2$^+$ at 3 weeks, although patches of cells which were down-regulating SOX9 were visible (bracket). By 7 and 12 weeks SOX9 was expressed at very low levels in some human cells. Green: SOX9; red: HuNu; white: SOX2. (E, F) Rare patches of differentiated human airway cells were identified in 2/3 organoid lines in kidneys harvested at 12 weeks. These were lined with basal, goblet and ciliated cells similar to the in vivo human airways. The remaining human airway-like structures contained goblet cells only and were found in every graft harvested (e.g. arrows in E and F). (E) Green: KRT5 (basal cells); red: Td-Tomato (mouse cells); white: MUC5AC (goblet cells). (F) Green: KRT5 (basal cells); red: Td-Tomato (mouse cells); white: ACT (cilia). Bars = 50 μm (C, D, E', F'); 2 mm (A); 0.5 mm (B); 200 μm (B'); 100 μm (E, F); 20 μm (E'', F'').

The following figure supplements are available for figure 5:

**Figure supplement 1.** Cells isolated from self-renewing organoids are competent to engraft into adult mouse lungs.

*Figure 5 continued on next page*

*Figure 5 continued*

**Figure supplement 2.** Mouse regions of chimeric human-mouse kidney capsule grafts differentiate efficiently.

cultures, we expanded fibroblasts from 19 or 20 pcw lungs and cultured with tip organoids. This resulted in more highly branched organoids with squamous epithelium surrounded by mesenchymal cells, reminiscent of the canalicular stage lung (*Figure 6I,J*). These organoids were SOX2⁻, SOX9⁻ and pro-SFTPC⁺, HOPX⁺; rare HTII-280⁺ cells were also observed (*Figure 6—figure supplement 4B–D*; n = 3 organoid lines), consistent with alveolar fate; most likely bipotential alveolar progenitors. When tip organoids derived from 19 pcw lungs were used there were extensive regions of pro-SFTPC, HTII-280, HOPX and PDPN co-expression and NKX2-1 was retained. This again indicates that we have successfully differentiated to the bipotent progenitor stage and also suggests that the organoids derived from the canalicular stage lungs are intrinsically easier to differentiate towards alveolar fate (*Figure 6H–K*; *Figure 6—figure supplemental 4 E–I* E-I; n = 2, 19 pcw organoid lines). Antibody staining showed that the expanded fibroblasts used in these experiments were PDGFRB⁺, but an otherwise heterogeneous mixture of cells expressing markers consistent with the mesenchyme observed in 20 pcw lung sections (*Figure 6—figure supplement 5*).

## Discussion

It is routinely assumed that the pseudoglandular stage human embryonic lung distal tip epithelium is a multipotent progenitor population. We now provide several lines of evidence that this hypothesis is correct. Firstly, we show that typical differentiation markers are detectable at a protein level only after cells exit the tip. Secondly, that 96% of human tip genes with mouse orthologues are also expressed in mouse tips. Thirdly, we demonstrate that the human tips have the ability to long-term self-renew and differentiate into bronchiolar and alveolar epithelium in vitro. Moreover, we have established conditions to grow human embryonic lung epithelial tips as long-term self-renewing organoids. These organoids retain many transcriptional similarities to tip cells and show no signs of spontaneous differentiation. They can also integrate into adult and embryonic lungs and be induced to differentiate towards bronchiolar or alveolar lineages in vitro. The organoid culture conditions that we have established provide a new tool for in vitro genetic studies of human lung development.

Although we find an extremely high level of transcriptome conservation between human and mouse tips we also report multiple differences, including in genes used as definitive cell-type specific markers. The most striking difference is in SOX2. Mouse tip progenitors are SOX2⁻, SOX9⁺ throughout development. By contrast, human pseudoglandular tips, which are producing bronchiolar descendants, are SOX2⁺, SOX9⁺ and tips become SOX2⁻, SOX9⁺ during the canalicular stage. The functional significance of tip SOX2 expression is currently unknown, although it may reflect lineage-priming. However, this observation is highly relevant to attempts to establish human pluripotent stem cell (PSC) differentiation protocols for lung epithelium. Based on mouse data, such human studies typically focus on the derivation of multipotent lung tip progenitors as *NKX2-1⁺*, *SOX2⁻*, *SOX9⁺*, *ID2⁺* cells. These are likely to be human canalicular tip progenitors. Differentiation marker expression in human PSC differentiation experiments is also based on mouse data. More robust, efficient protocols for human PSC differentiation are likely to be developed if human lung development is taken into account.

We have observed that mouse tip progenitors are unable to long-term self-renew in human culture conditions, even though mouse-specific versions of the factors were provided. Hence, the tip transcriptome is highly conserved between mouse and human, but the differences are of functional significance. There is therefore a strong imperative to study human lung development alongside more traditional mouse studies. We have used organoid culture to show that differentiating human lung stalks can be reprogrammed to tip fate analogous to classical rodent studies (*Alescio and Cassini, 1962*); FGF and WNT are required to maintain human tip *SOX2* and *SOX9*; and activation of EGF, FGF and WNT, with inhibition of BMP and TGFβ, signalling is sufficient to maintain human tip self-renewal. Moreover, our long-term self-renewing organoids are competent to differentiate, although in vitro differentiation requires further optimisation. We have also introduced plasmids to

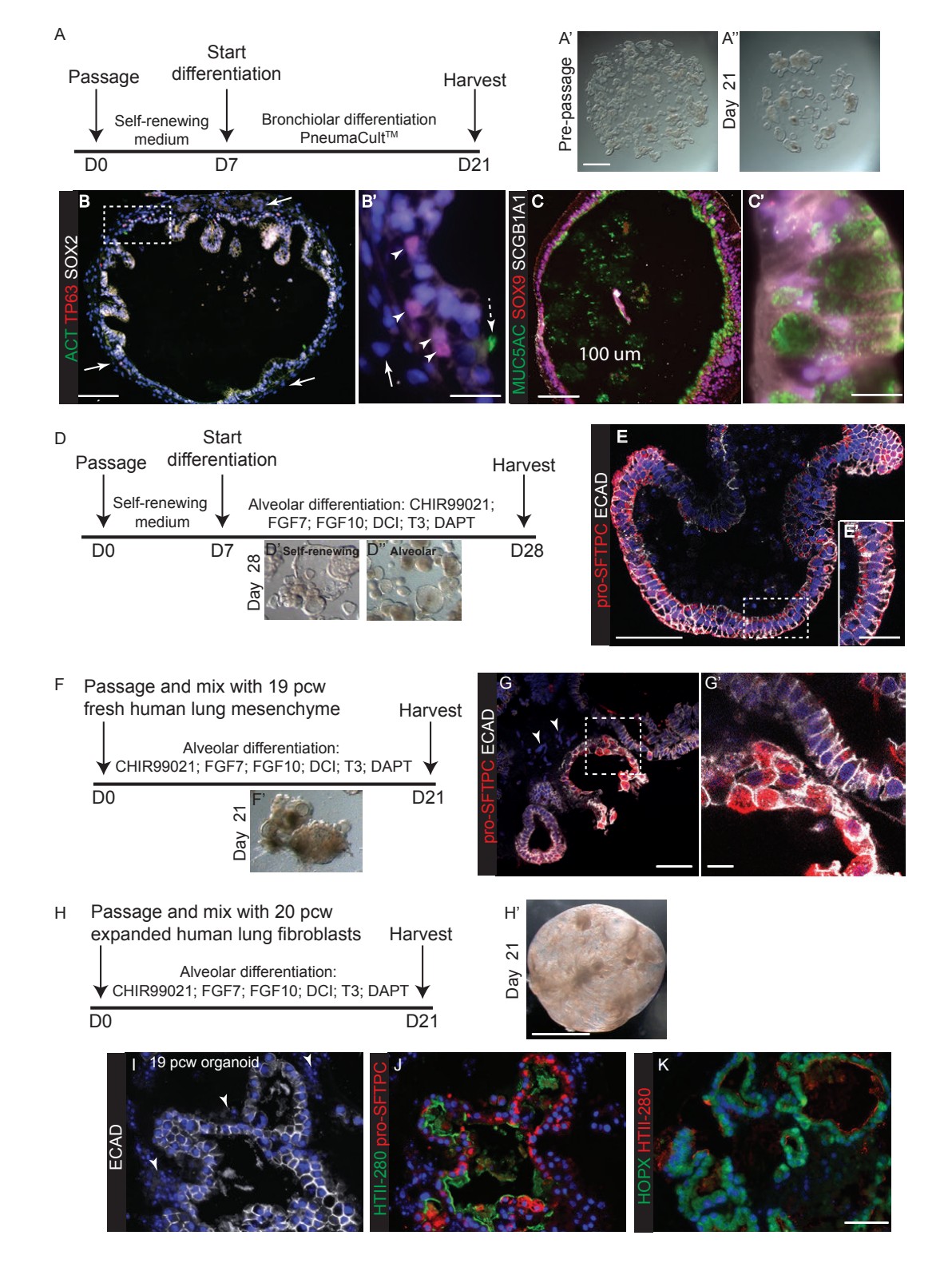

**Figure 6.** In vitro differentiation of self-renewing organoids towards bronchiolar and alveolar lineages. (**A**) Experimental schematic for 3 weeks organoid differentiation in Pneumacult-ALI medium. (**B**) In low passage number organoids, mesenchyme expanded (arrows), basally-located TP63⁺ basal cells differentiated and rare ACT⁺ ciliated cells were seen. Dashed arrow = ciliated cell. Arrow heads = basal cells. Arrow = mesenchyme. Green: ACT (cilia); red: TP63 (basal cells); white: SOX2. (**C**) Differentiated organoids were predominantly composed of MUC5AC⁺ goblet cells, although low levels of

*Figure 6 continued on next page*

*Figure 6 continued*

SCGB1A1 were observed in some cells. Cells retained SOX9 suggesting that differentiation was not complete. Green: MUC5AC (mucous); red: SOX9; white: SCGB1A1. (D) Experimental schematic for 3 week alveolar differentiation experiment. (E) All organoid cells retained a columnar appearance and expressed relatively low levels of apical pro-SFTPC. Red: pro-SFTPC; white: ECAD. (F) Experimental schematic for 3 week alveolar differentiation in the presence of freshly-isolated 19 pcw human mesenchyme. (G) Mesenchymal cells were observed in the cultures (arrowheads), moreover cells expressing higher levels of pro-SFTPC with a more squamous appearance were also obtained. Red: pro-SFTPC; white: ECAD. (H) Experimental schematic for 3 week alveolar differentiation of 19 pcw organoid in the presence of expanded human 20 pcw mesenchymal cells. DCI = dexamethasone, cAMP, IBMX. (I) Organoid epithelium took on a more squamous appearance and was surrounded by mesenchymal cells (arrowheads). White: ECAD. (J) AT2 markers were expressed robustly. Green: HTII-280; red: pro-SFTPC. (K) AT1 and AT2 markers were co-expressed. Green: HOPX; red: pro-SFTPC. Blue: DAPI. Bars = 1 mm (A, H); 100 μm (B, C); 20 μm (B′, E′); 10 μm (C′, G′); 50 μm (E, G, I, J, K).

The following figure supplements are available for figure 6:

**Figure supplement 1.** Exposure of self-renewing organoids to Pneumacult™ medium leads to efficient goblet cell differentiation with rare patches of KRT5[+] basal cells.

**Figure supplement 2.** Testing media conditions for ability to promote human organoid alveolar differentiation.

**Figure supplement 3.** A combination of canalicular stage lung embryonic mesenchyme and alveolar differentiation medium together promote the most efficient organoid alveolar differentiation.

**Figure supplement 4.** Expanded canalicular stage mesenchyme and alveolar medium can promote organoid alveolar differentiation.

**Figure supplement 5.** Expanded fibroblasts used for organoid co-cultures are a heterogeneous population expressing various lung embryonic fibroblast markers.

the self-renewing organoids by electroporation (*Fujii et al., 2015*) and been able to freeze-thaw organoids for longer-term storage. Therefore, a genetic system for the study of human lung development is now available. We propose that this will also be useful for disease modelling and informing the differentiation of human PSCs.

## Materials and methods

### Human embryonic and foetal lung tissue

Human embryonic and foetal lungs were obtained from terminations of pregnancy from Cambridge University Hospitals NHS Foundation Trust under permission from NHS Research Ethical Committee (96/085) and the Joint MRC/Wellcome Trust Human Developmental Biology Resource (London and Newcastle, grant 099175/Z/12/Z, www.hdbr.org). Their age ranged from 5 to 20 weeks developmental age, also known as post-conception weeks, pcw (this corresponds to 7–22 weeks gestational age). Samples were staged according to their external physical appearance and measurements, and not to the estimated last menstrual period. Detailed guidelines for embryonic samples (<8 pcw): http://hdbr.org/downloads/embryo_staging_guidelines.doc; and for foetal samples: http://hdbr.org/downloads/fetal_staging_guidlines.doc. Samples used had no known genetic abnormalities.

### Human adult lung tissue

Fresh healthy adult lung tissue (background tissue from lobectomies for lung cancer) was obtained from Papworth Hospital NHS Foundation Trust (Research Tissue Bank Generic REC approval, Tissue Bank Project number T01939) and processed for both cryo- and paraffin sectioning.

### Animals

All experiments were approved by local ethical review committees and conducted according to Home Office project licenses PPL 70/8012 (Emma Rawlins, University of Cambridge) and 70/7607 (Adam Giangreco, UCL). Mouse strains (*Rosa26R-mT/mG*, formally known as *Gt(ROSA)26Sor^{tm4 (ACTB-tdTomato,-EGFP)Luo}*/J; RRID:IMSR_JAX:026862) (*Muzumdar et al., 2007*) and *NOD-scid-IL2rg^{-/-}*

(NSG; RRID:IMSR_JAX:005557) (*Ishikawa et al., 2005*; *Shultz et al., 2005*) have been described. Wild-type mice were outbred MF1 strain.

## Dissection of human and mouse embryonic lungs and set-up of organoid culture

Lung lobes were incubated for 2 min in Dispase (ThermoFisher Scientific, Gibco, UK, 8 U/ml) at room temperature. Mesenchyme was dissected away using tungsten needles and tips and stalks were isolated by cutting the very end of a branching tip, or alternatively a stalk area more proximally. Five tips, or 2 pieces of stalk tissue, were transferred into 30 µl Matrigel (Corning, UK, 356231). A dissecting microscope was used to guide aspiration of 25 µl Matrigel containing the tissue pieces, which was transferred into a well of a 48 well low-attachment plate (Greiner, UK). The plate was incubated for 5 min at 37°C to solidify the Matrigel, following which at least 250 µl of self-renewing was added (*Table 1*, or *Table 2*). Plates were incubated under standard tissue culture conditions (37°C, 5% $CO_2$).

## Growing human foetal lung mesenchyme

92 mm plates were coated with type I collagen (Sigma, UK, C3867-1VL) mixed with 0.02 N acetic acid (1:72.5) using a total volume of 6 ml, then left to evaporate in a tissue culture hood for about 4 hr. Fresh human foetal lung was cut into small pieces and incubated at 37°C, 30 min in 24 U/ml Dispase (Gibco), 10 µg/ml DNase in PBS. DMEM/F12 with 10% (v/v) FBS (ThermoFisher Scientific, Life Technologies) was added and lung pieces spun 200 g, 5 min. The supernatant was aspirated and the pellet was resuspended in DMEM/F12 with 10% (v/v) FBS and 1:100 Penicillin/Streptomycin (ThermoFisher Scientific, Life Technologies). The lung pieces and culture medium were transferred evenly onto the collagen coated plate. The plate was incubated for 5 days without medium change. On day 5, the lung pieces were removed and fresh medium added. Medium change was twice a week until confluence. Cells were split using 0.1% (w/v) trypsin for 2 min at 37°C, inactivated with

**Table 1.** *Self-renewal (Human)*

| Reagent | Company | Cat no | Final concentration |
| --- | --- | --- | --- |
| Advanced DMEM | ThermoFisher Scientific, Invitrogen | 12634–010 | Base medium |
| Penicillin/Streptomycin | ThermoFisher Scientific, Invitrogen | 15140–122 | 100 U/ml (Pen) 100 µg/ml (Strep) |
| Hepes | ThermoFisher Scientific, Invitrogen | 15630–056 | 10 mM |
| Glutamax | ThermoFisher Scientific, Invitrogen | 35050–038 | 2 mM |
| N2 | ThermoFisher Scientific, Invitrogen | 17502–048 | 1:100 |
| B27 (-Vit A) | ThermoFisher Scientific, Invitrogen | 12587–010 | 1:50 |
| N-acetylcysteine | Sigma-Aldrich | A9165 | 1.25 mM |
| Matrigel (growth factor reduced; specific lots of matrigel with at least 8 mg/ml protein concentration were used) | Corning | 356231 | undiluted |
| R-spondin1 conditioned medium | Stem Cell Intitute, University of Cambridge | From 293T-HA-Rspo1-Fc cell line made by Calvin Kuo, Stanford | 5% v/v |
| EGF | Peprotech, UK | AF-100–15 | 50 ng/ml |
| Noggin | R and D Systems | 6057 NG-100 | 100 ng/ml |
| FGF10 | R and D Systems | 345-FG-025 | 100 ng/ml |
| FGF7 | Peprotech | 100–19 | 100 ng/ml |
| CHIR 99021 | Stem Cell Institute, University of Cambridge | n/a | 3 µM |
| SB 431542 | Tocris | 1614 | 10 µM |
| 48 well plates (Greiner Cellstar) | Sigma-Aldrich | M9437 | n/a |

**Table 2.** *Self-renewal (Mouse)*

| Reagent | Company | Cat no | Final concentration |
|---------|---------|--------|---------------------|
| Advanced DMEM | ThermoFisher Scientific, Invitrogen | 12634–010 | n/a |
| Penicillin/Streptomycin | ThermoFisher Scientific, Invitrogen | 15140–122 | 100 U/ml (Pen) 100 µg/ml (Strep) |
| Hepes | ThermoFisher Scientific, Invitrogen | 15630–056 | 10 mM |
| Glutamax | ThermoFisher Scientific, Invitrogen | 35050–038 | 2 mM |
| N2 | ThermoFisher Scientific, Invitrogen | 17502–048 | 1:100 |
| B27 (-Vit A) | ThermoFisher Scientific, Invitrogen | 12587–010 | 1:50 |
| N-acetylcysteine | Sigma-Aldrich | A9165 | 1.25 mM |
| Matrigel | Corning | 356231 | Undiluted |
| R-spondin conditioned medium | Stem Cell Institute, University of Cambridge | From 293T-HA-Rspo1-Fc cell line made by Calvin Kuo, Stanford | 5% v/v |
| mEGF | R and D Systems | 2028-EG-200 | 50 ng/ml |
| Noggin | R and D Systems | 6057 NG-100 | 100 ng/ml |
| FGF10 | R and D Systems | 345-FG-025 | 100 ng/ml |
| mFGF7 | R and D Systems | 5028 KG_025 | 100 ng/ml |
| CHIR 99021 | Stem Cell Institute, University of Cambridge | n/a | 3 µM |
| SB 431542 | Tocris | 1614 | 10 µM |

DMEM/F12 with 10% (v/v) FBS, centrifuged and then plated on 92 mm plates. For antibody staining fibroblasts were passaged onto collagen-coated coverslips. (Consumable details, *Table 3*)

## Maintenance of 3D human and mouse embryonic lung organoid culture

Organoids were cultured in Matrigel (Corning, 356231) in 48-well plates with self-renewing medium (Advanced DMEM/F12 supplemented with 1x GlutaMax, 1 mM Hepes and Penicillin/ Streptomycin (P/S), 1:50 B27 supplement (without Vitamin A), 1:100 N2 supplement, 1.25 mM n-Acetylcysteine, 5% (v/v) R-spondin1 conditioned medium, 50 ng/ml recombinant human EGF, 100 ng/ml recombinant human Noggin, 100 ng/ml recombinant human FGF10, 100 ng/ml recombinant human FGF7, 3 µM CHIR99021 and 10 µM SB431542 (*Table 1*). For mouse cultures, mouse specific EGF and FGF7 were used (*Table 2*). Medium was changed twice a week, and organoids were passaged every 10–14 days depending on cell confluence and Matrigel stability. Plates were incubated under standard

**Table 3.** *Human foetal lung mesenchyme*

| Reagent | Company | Cat no | Final concentration |
|---------|---------|--------|---------------------|
| Dish Nunc T/C 92 mm | ThermoFisher Scientific | 10508921 | n/a |
| Type I collagen | Sigma-Aldrich | C3867-1VL | 55 µg/ml |
| Acetic acid | ThermoFisher Scientific | 10304980 | 0.02 N |
| DMEM/F12 with L-glutamine | ThermoFisher Scientific, Invitrogen | 11320–074 | n/a |
| Dispase | ThermoFisher Scientific, Invitrogen | 17105041 | 24 U/ml |
| FBS | Sigma-Aldrich | F9665 | 10% |
| Penicillin/Streptomycin | ThermoFisher Scientific, Invitrogen | 15140–122 | 100 U/ml (Pen) 100 µg/ml (Strep) |
| DNase I | Qiagen, UK | 79254 | 10 µg/ml |
| Trypsin (from porcine pancreas) | Sigma-Aldrich | T4799 | 0.1% |

tissue culture conditions (37°C, 5% $CO_2$). Unless otherwise stated organoid lines were used between passage 4 and passage 16 for experiments.

## Passaging of organoids

Organoids were usually split 1:4 to 1:6 after 10–14 days of culture. The medium was aspirated and fresh cold base medium: Advanced DMEM with Glutamax, P/S and Hepes (AdvDMEM+++) added to each well. The Matrigel in each well was sucked into a P1000 pipette tip and transferred into a 15 ml tube. Cold AdvDMEM+++ was added up to 10 ml and then the sample was centrifuged at 100 g at 4°C for 5 min. 8.5 ml of medium was then aspirated, and the remaining organoids triturated using a flame polished glass pipette. Cold AdvDMEM+++ was added up to 10 ml and the sample was again centrifuged at 220 g at 4°C. The pellet was resuspended in undiluted Matrigel (in a volume depending on the splitting ratio) and 25 µl of Matrigel containing the split organoids was plated onto a well of a 48 well low attachment plate. The plate was incubated for 5 min at 37°C to allow the Matrigel to solidify, upon which at least 250 µl culture medium was added per well (**Table 1** or **Table 2**). For bronchiolar or alveolar differentiation different media were used as outlined in the figures (**Table 4**, **Table 5**). Organoids, and other primary cells, were tested regularly for mycoplasma.

Organoid lines could be frozen and thawed without apparent change in behaviour. For freezing, organoids were removed from matrigel and triturated with a flame polished glass pipette as for passaging. They were then pelleted and resuspended in cold freezing medium (Invitrogen, 12648010) at 500 µl per well in a cryovial which was transferred into a pre-cooled Mr. Frosty Freezing Container (ThermoFisher Scientific, Invitrogen, 5100–0001) at −80°C overnight followed by longer-term storage in liquid nitrogen. Cryovials were thawed for 2 min in a 37°C waterbath and organoids plated in matrigel in self-renewing medium supplemented with 10 µl Rho kinase inhibitor (Y27632, Sigma-Aldrich, Y0503-1MG).

When fresh, or expanded, human mesenchyme was added to the cultures, 250, 000 mesenchymal cells per well were mixed with the organoids immediately prior to the final spin and resuspension in Matrigel. All differentiation experiments were performed in at least three technical replicates.

## Recovering organoids from matrigel for immunostaining or RNA extraction

Prior to fixation, or lysis, organoids were removed from Matrigel using Corning Matrigel Cell Recovery Solution (Corning, 354253). First, organoids were harvested into a 15 ml tube using a wide Pasteur pipette and washed with 10 ml of cold washing medium (Advanced DMEM/F12, 1X GlutaMax, 1 mM Hepes and Penicillin/Streptomycin). The 15 ml tube was inverted every 2 min for 10 min, followed by 5 min incubation on ice before organoids were spun 200 g at 4°C. This was repeated once and then Corning Cell Recovery Solution (Corning, 354253) was used to further remove the Matrigel (incubation on ice for 30 min with inversion once after 15 min). Organoids were washed with cold PBS, spun down at 200 g 4°C.

## Whole mount immunostaining of organoids and embryonic lungs

For 5–9 pcw lungs fixation was overnight hour at 4°C in 4% PFA. Organoids were recovered from the Matrigel using Corning Cell Recovery Solution (Corning, 354253) as above and fixed 4% (w/v) paraformaldehyde (PFA) for 30 min at 4°C. After washing in PBS organoids were transferred to a round-bottom 96 well plate using wide Pasteur pipettes. Permeabilisation in 0.5% (v/v) Triton-X in PBS for 30 min was followed by washing in 0.5% (w/v) Bovine Serum Albumin (BSA), 0.2% Triton-X in PBS (washing solution). Blocking was for at least 1 hr at room temperature in 1% BSA, 5% NDS (normal donkey serum), 0.2% Triton-X in PBS. Primary antibodies (**Table 6**) in blocking solution used at

**Table 4.** *Human Bronchiolar differentiation*

| Reagent | Company | Cat no | Final concentration |
|---|---|---|---|
| PneumaCult™-ALI medium | Stem Cell Technologies | 05001 | n/a |
| Matrigel | Corning/SLS | 356231 | undiluted |

**Table 5.** *Human Alveolar differentiation*

| Reagent | Company | Cat no | Final concentration |
|---|---|---|---|
| Advanced DMEM | ThermoFisher Scientific, Invitrogen | 12634–010 | n/a |
| Penicillin/Streptomycin | ThermoFisher Scientific, Invitrogen | 15140–122 | 100 U/ml (Pen) 100 µg/ml (Strep) |
| Hepes | ThermoFisher Scientific, Invitrogen | 15630–056 | 10 mM |
| Glutamax | ThermoFisher Scientific, Invitrogen | 35050–038 | 2 mM |
| N2 | ThermoFisher Scientific, Invitrogen | 17502–048 | 1:100 |
| B27 (-Vit A) | ThermoFisher Scientific, Invitrogen | 12587–010 | 1:50 |
| N-acetylcysteine | Sigma-Aldrich | A9165 | 1.25 mM |
| Matrigel | Corning/SLS | 356231 | Undiluted |
| CHIR 99021 | Stem Cell Institute, University of Cambridge | n/a | 1 µM or 3 µM |
| FGF10 | R and D | 345-FG-025 | 100 ng/ml |
| FGF7 | Peprotech | 100–19 | 100 ng/ml |
| Dexamethasone | Sigma-Aldrich | D4902-25MG | 50 nM |
| cAMP | Sigma-Aldrich | B5386-5MG | 0.1 mM |
| IBMX | Sigma-Aldrich | I5879-100MG | 0.1 mM |
| DAPT | Sigma-Aldrich | D5942-25MG | 50 µM |
| Tri-iodothyronine (T3) | Sigma-Aldrich | T6397-100MG | 6.7 ng/ml |
| Human IGF-1 | R and D Systems | 291-G1-200 | 1 µg/ml |
| Interleukin-6 | R and D Systems | 206-IL-010 | 1 µg/ml |

4°C overnight. The following day washes were performed at 4°C and secondary antibodies (1:2000 dilution; *Table 7*) in 5% NDS, 0.2% Triton-X in PBS incubated overnight at 4°C. The following day washes were performed at 4°C and DAPI (Sigma) added to the washing solution for 30 min at 4°C. Samples were processed to 2′−2′-thio-diethanol (TDE, Sigma, 166782) for clearing/mounting: 10% (v/v) TDE in 1x PBS; 25%; 50% 1 hr, 97% TDE overnight at 4°C on a rocker. The following day, organoids were transferred onto a slide with an imaging spacer (diameter 20 mm; thickness 0.12 mm; Sigma GBL654006) containing 65 µl 97% (v/v) TDE and coverslipped.

## Immunohistochemistry (cryosections)

Mouse embryonic lungs were fixed at 4°C in 4% PFA for 1 hr. Fixation was at 4°C in 4% PFA overnight for mouse kidneys and human embryonic, foetal and adult lungs; after nine pcw the lungs were divided into pieces prior to fixation, preferably intact lung lobes, in order to fit in 15x15 × 5 mm moulds. Post-fixation PBS washes and sucrose protection (15%, 20%, 30% w/v sucrose in PBS 1 hr each) were at room temperature. Samples were incubated 1:1 in 30% sucrose: optimal cutting temperature compound (OCT) overnight at 4°C, 1 hr room temperature 100% OCT wash for small tissue fragments only, then embedded in OCT. 7 µm sections were cut and stored at −80°C. Tissue was permeabilised using 0.3% Triton-X in PBS. Antigen retrieval was by heating slides in 10 mM Na Citrate buffer at pH6 in a full power microwave for 5 min. Blocking at least 1 hr, room temperature in 5% NDS, 1% BSA, 0.1% Triton-X in PBS. Primary antibodies (*Table 6*) were diluted in block and incubated overnight 4°C. After PBS washes, secondary antibodies (1:2000; *Table 7*) were added in 5% NDS, 0.1% Triton-X in PBS and incubated 2–3 hr at room temperature. When biotin-coupled secondaries were used, sections were incubated in 1.8 µg/ml Streptavidin-594 in 1% BSA for 30 min at room temperature. DAPI (Sigma) was added for 20 min, followed by PBS washes and mounting in Fluoromount (Sigma). Images were collected on a Zeiss Axiophot microscope, or Leica SP8 confocal where stated.

For sections of human lungs, for each antibody at least 3 different 5–8 pcw lungs; 2 different 11 pcw lungs; 2 different 14–15 pcw lungs; 2 different 16–17 pcw lungs; one 19 pcw lung and 2

**Table 6.** *Primary antibodies*

| Antibody | Company | Cat no | Host species | Final dilution | Antigen retrieval (Cryo) | Antigen retrieval (Paraffin) | Research Resource Identifier (RRID) |
|---|---|---|---|---|---|---|---|
| ABCA3 | Seven Hills Bioreagents, Cincinnati, OH | WRAB-ABCA3 | Rabbit | 1:500 | No | Yes | RRID:AB_577286 |
| Acetylated tubulin (ACT) | Sigma | T7451, clone 6-11B-1 | Mouse | 1:3000 | No | No | RRID:AB_609894 |
| AQP5 | Santa Cruz Biotechnology, Dallas, Texas | Sc9890, clone G19 | Goat | 1:100 | Yes | Yes | RRID:AB_2059877 |
| β3-TUBULIN | Biolegend (Covance), UK | PRB-435P | Rabbit | 1:1000 | No | N/A | RRID:AB_291637 |
| CD90 | Novus Biologicals, UK | NBP2-37330 (clone 7E1B11) | Mouse | 1:200 | No | N/A | RRID:AB_2665376 |
| E-CADHERIN | ThermoFisher Scientific Invitrogen | 13–1900 | Rat | 1:3000 | No | N/A | RRID:AB_2533005 |
| E-CADHERIN | BD Biosciences, UK | 610182 | Mouse | 1:500 | No | Yes (citrate) | RRID:AB_397581 |
| ETV5 | Santa Cruz Biotechnology | Sc-22807 | Rabbit | 1:200 | Yes | N/A | RRID:AB_2101008 |
| FGFR2 | Santa Cruz Biotechnology | SC-122 | Rabbit | 1:200 | No | N/A | RRID:AB_631509 |
| FOXA2 | Santa Cruz Biotechnology | SC-6554 | Goat | 1:200 | No | N/A | RRID:AB_2262810 |
| FOXJ1 | Thermo Fisher Scientific | 14-9965-82 | Mouse | 1:200 | Yes | Yes (citrate); needs streptavidin-biotin amplification | RRID:AB_1548835 |
| HMGA1B | R and D Systems, UK | AF5956 | Sheep | 1:50 | Yes | N/A | RRID:AB_1964602 |
| HMGA2 | Proteintech, UK | 20795–1-AP | Rabbit | 1:100 | No | N/A | RRID:AB_2665377 |
| HNF1B | Proteintech | 12533–1-AP | Rabbit | 1:100 | Yes | N/A | RRID:AB_2116758 |
| HOPX | Santa Cruz Biotechnology | SC-30216 | Rabbit | 1:50 | No | Yes (citrate) | RRID:AB_2120833 |
| HTI-56 | Gift from Leland Dobbs | n/a | Mouse | 1:100 | No | N/A | RRID:AB_2665380 |
| HTII-280 | Gift from Leland Dobbs | n/a | Mouse IgM | 1:100 | No | No | RRID:AB_2665381 |
| Human Nuclei (HuNu) | Merck, UK | MAB1281 | Mouse | 1:3000 | No (needs streptavidin-biotin amplification) | N/A | RRID:AB_11212527 |
| ID2 | Abcam, UK | Ab52093 | Rabbit | 1:200 | Yes | N/A | RRID:AB_880731 |
| KRT5 | Covance | PRB-160P-100 | Rabbit | 1:500 | No | Yes (citrate) | RRID:AB_291581 |
| KI67 | BD Transduction Laboratories, UK | 550609, clone B56 | Mouse | 1:100 | No | Yes (citrate) | RRID:AB_393778 |
| LPCAT1 | Proteintech | 16112–1-AP | Rabbit | 1:500 | No | Yes (citrate) | RRID:AB_2135554 |
| MUC5AC | ThermoFisher Scientific | MS-145P | Mouse | 1:500 | No | Yes (citrate) | RRID:AB_62731 |
| NKX2-1 | Abcam | Ab76013 | Rabbit | 1:500 | Yes | Yes (citrate) | RRID: AB_1310784 |
| PDGFRA | Cell Signalling | 3174 (clone D1E1E) | Rabbit | 1:1000 | No | N/A | RRID:AB_2162345 |
| PDGFRB | Cell Signalling | 3169 (clone 28E1) | Rabbit | 1:100 | No | N/A | RRID:AB_2162497 |
| PDPN | Proteintech | 11629–1-AP | Rabbit | 1:200 | No | Yes | RRID:AB_2162067 |
| PDPN | R and D Systems | AF3670 | Sheep | 1:200 | No | Yes | RRID:AB_2162070 |
| PECAM (CD31) | Abcam | Ab9498 | Mouse | 1:200 | No | N/A | RRID:AB_307284 |
| SCGB1A | Santa Cruz Biotechnology | SC-25555 | Rabbit | 1:200 | No | Yes (citrate or trypsin) | RRID:AB_2269914 |
| pro-SFTPC | Millipore, UK | Ab3786 | Rabbit | 1:500 | No | Yes (citrate) | RRID:AB_91588 |

*Table 6 continued on next page*

*Table 6 continued*

| Antibody | Company | Cat no | Host species | Final dilution | Antigen retrieval (Cryo) | Antigen retrieval (Paraffin) | Research Resource Identifier (RRID) |
|---|---|---|---|---|---|---|---|
| SMA | Sigma | A5228, clone 1A4 | Mouse | 1:500 | No | N/A | RRID:AB_262054 |
| SOX2 | Santa Cruz Biotechnology | SC-17320 | Goat | 1:250 | No/Yes | Yes (citrate) | RRID:AB_2286684 |
| SOX9 | Santa Cruz Biotechnology | SC-20095 | Rabbit | 1:200 | No | Yes (citrate) | RRID:AB_661282 |
| SOX9 | Abcam | ab196450 | Rabbit | 1:200 | No | N/A | RRID:AB_2665383 |
| SPRY2 | Abcam | ab50317 | Rabbit | 1:200 | Yes | N/A | RRID:AB_882688 |
| TP63 | Cell Signaling | 13109 | Rabbit | 1:200 | Yes | Yes (citrate); needs streptavidin-biotin amplification | RRID:AB_2637091 |
| VECAD | R and D Systems | AF938 | Goat | 1:400 | No | N/A | RRID:AB_355726 |

different 20 pcw lungs were stained. At least two technical replicates were performed for each immunostaining.

## Immunohistochemistry (paraffin)

Human embryonic and adult lungs were cut to fit 15x15 × 5 mm moulds and fixed at 4°C in 4% (w/v) PFA overnight. Organoids were removed from the Matrigel following the Corning Cell Recovery Solution protocol (above) and fixed for 30 min at 4°C. Following PBS washes, small tissue (e.g. organoids) was embedded in 3% (w/v) Low Melting Point Agarose (Sigma, A2790). All samples were dehydrated to 100% ethanol and then processed to paraffin wax for embedding. Paraffin blocks were sectioned at 5 µm and slides dried at 50°C for at least 30 min. Deparaffinisation was performed by 2x xylene washes, followed by rehydration to distilled water then PBS rinse. Antigen retrieval was heating in 10 mM Na Citrate buffer at pH6, or 0.05% (w/v) trypsin in PBS as appropriate. Blocking at least 1 hr, room temperature in 5% NDS, 1% BSA, 0.1% Triton-X in PBS. Primary antibodies (*Table 6*) were diluted in block and incubated overnight 4°C. After PBS washes, secondary antibodies (1:2000; *Table 7*) were added in 5% NDS, 0.1% Triton-X in PBS and incubated 2–3 hr at room temperature. When biotin-coupled secondaries were used, sections were incubated in 1.8 µg/ml Streptavidin-594 in 1% BSA for 30 min at room temperature. DAPI (Sigma) was added for 20 min, followed by PBS

**Table 7.** *Secondary antibodies*

| Antibody | Company | Cat no | Final dilution | Research Resource Identifier (RRID) |
|---|---|---|---|---|
| Donkey α-mouse 488 | Thermo Fisher Scientific | A21202 | 1:2000 | RRID:AB_141607 |
| Donkey α-rabbit 488 | Thermo Fisher Scientific | A21206 | 1:2000 | RRID:AB_2535792 |
| Donkey α-rat 488 | Thermo Fisher Scientific | A21208 | 1:2000 | RRID:AB_2535794 |
| Donkey α-mouse 546 | Thermo Fisher Scientific | A10036 | 1:2000 | RRID:AB_2534012 |
| Donkey α-rabbit 546 | Thermo Fisher Scientific | A10040 | 1:2000 | RRID:AB_2534016 |
| Donkey α-goat 555 | Thermo Fisher Scientific | A21432 | 1:2000 | RRID:AB_2535853 |
| Donkey α-rat 594 | Thermo Fisher Scientific | A21209 | 1:2000 | RRID:AB_2535795 |
| Donkey α-sheep 594 | Jackson Immunoresearch, West Grove, PA | 713-585-147 | 1:2000 | RRID:AB_2340748 |
| Donkey α-mouse 647 | Thermo Fisher Scientific | A31571 | 1:2000 | RRID:AB_162542 |
| Donkey α-rabbit 647 | Thermo Fisher Scientific | A31573 | 1:2000 | RRID:AB_2536183 |
| Goat α-rat 647 | Thermo Fisher Scientific | A21247 | 1:2000 | RRID:AB_141778 |
| Streptavidin 594 | Jackson Immunoresearch | 016-580-084 | 1.8 µg/ml | RRID:AB_2337250 |
| Biotin-SP-conjugated Donkey α-mouse | Jackson Immunoresearch | 715-065-150 | 1:500 | RRID:AB_2307438 |

washes and mounting in Fluoromount (Sigma). Images were collected on a Zeiss Axiophot microscope, or Leica SP8 confocal where stated. Haematoxylin and Eosin staining followed standard protocols.

### RNA extraction, cDNA synthesis and qRT-PCR analysis

Organoids were removed from the Matrigel following the Corning Cell Recovery Solution protocol (above) and lysed using 500 μl RLT buffer. Freshly isolated tips and stalks were lysed using 350 μl RLT buffer. RNA extraction was performed according to the RNeasy Mini Kit protocol (Qiagen, UK). RNA concentrations were measured using Nanodrop (ThermoFisher Scientific). First Strand cDNA synthesis was performed using 1 μg RNA and the Superscript III RT system (ThermoFisher Scientific). cDNA was diluted 1:10 and 2–4 μl was used for one qPCR reaction with Taqman assays (ThermoFisher Scientific; *Table 8*). Relative gene expression was calculated using the *ΔΔ*CT method relative to GAPDH control. P-values were obtained using an unpaired two-tailed student's t-test with unequal variance.

### RNA-seq sample preparation

Four age-matched biological replicates (6–7 pcw) were selected based on size and hand/foot morphology. Fresh tips and stalks were microdissected and cleaned of mesenchyme using tungsten needles following 2 min in Dispase (Gibco, 16 U/ml final concentration) at room temperature. Microdissected tips (~8) and stalks (~8) were transferred by mouth pipette into 50 μl extraction buffer using the PicoPure RNA Isolation Kit (ThermoFisher Scientific) in DNA LoBind tubes (Eppendorf). Organoids were removed from the Matrigel following the Corning Cell Recovery Solution protocol (above) and transferred into 50 μl extraction buffer. RNA extraction was performed according to the PicoPure RNA Isolation Kit protocol. Total RNA concentration and quality using RIN score was assessed using RNA 6000 Pico Kit (Agilent). Only biological replicates with RIN score >8 were used. Reverse transcription and cDNA amplification was performed according to Ovation RNA-Seq Systems V2 protocol (NuGEN). For each sample a minimum total input RNA amount of 500 pg was used. Quality check was performed using the Agilent DNA 1000 kit and RNA-Seq library preparation was performed according to the NuGEN Ovation Rapid DR Multiplex System 1–8 protocol. Sequencing was performed at the Gurdon Institute on a HiSeq 1500 in rapid run mode (Illumina, San Diego, CA; single read 50 nucleotides). All RNAseq data deposited in GEO: https://www.ncbi.nlm.nih.gov/geo/query/acc.cgi?token=chyfeyegvxurngn&acc=GSE95860.

### Bioinformatics analysis

For comparing freshly isolated tips to freshly isolated stalks (*Figure 2*), Fastq files were filtered for low quality reads (<Q20) and low quality bases were trimmed from the ends of the reads (<Q20) using Sickle. The resulting reads were mapped to the human reference genome (UCSC GRCh37/hg19) using TopHat 2.0.6 (*Kim et al., 2013*) guided by RefSeq gene models (UCSC). Raw counts per transcripts were obtained using *featureCounts* and differentially expressed genes (>2 fold difference) identified using edgeR 2.6.12 (*Robinson et al., 2010*). Hierarchical unsupervised clustering was performed using published foetal lung RNAseq data as a comparison (*Table 9*). Gene Ontology and Panther Pathway analysis was performed in DAVID (*Huang et al., 2009a*, *2009b*).

To compare the human and mouse embryonic tip transcriptome, we compared our human tip RNA seq with previously published mouse E11.5 tip microarrays (GEO accession numbers: GSM1968996, GSM1968997, GSM1968998, GSM1968999, GSM1969000). We first assessed whether

**Table 8.** *RT-PCR primers*

| TaqMan primer | Company | Cat no | Id no |
| --- | --- | --- | --- |
| GAPDH | Thermo Fisher Scientific | # 4326317E | n/a |
| SOX2 | Thermo Fisher Scientific | # 4331182 | Hs01053049_s1 |
| SOX9 | Thermo Fisher Scientific | # 4331182 | Hs01001343_g1 |
| TBX4 | Thermo Fisher Scientific | # 4331182 | Hs00218515_m1 |

**Table 9.** *Published whole foetal lung RNAseq (**Bernstein et al., 2010**).*

| Sample label | Description | Source | Donor ID | Accession # |
|---|---|---|---|---|
| Lung_A_6 | Foetal day 105 | GEO Datasets | H-24005 | GSM1101693 |
| Lung_F_2 | Foetal day 105 | GEO Datasets | H-24111 | GSM1101708 |
| Lung_F_3 | Foetal day 108 | GEO Datasets | H-23887 | GSM1101684 |
| Lung_F_4 | Foetal day 91 | GEO Datasets | H-23914 | GSM1101685 |
| Lung_F_5 | Foetal day 96 | GEO Datasets | H-24089 | GSM1101699 |
| Lung_F_6 | Foetal day 98 | GEO Datasets | H-23964 | GSM1101687 |

transcripts of orthologous mouse/human genes (defined by the HomoloGene database) were present in each data-set (*Figure 2—figure supplement 3A*). Genes were excluded which had RPKM values < 1 (RNAseq) and expression values of <5 (microarray). To estimate the relative levels of these transcripts between the mouse microarray and human RNAseq data, we reasoned that the microarray signal saturates and therefore generated a scatter plot of mean microarray signal, versus mean log-transformed RPKM for each orthologous gene identified in mouse and human (*Figure 2—figure supplement 3B*).

RNAseq for the cultured organoids was performed on an independent sequencing run. To compare freshly isolated human tip and stalk samples with the cultured organoids, the RUVSeq R package (ruvg using housekeeping genes) was used to control for the batch effect in the data (*Risso et al., 2014*) using hidden factor k = 1. Multi-dimensional scaling plot, heat map and box plots (*Figure 3* and *Figure 3—figure supplement 3*) were produced in R using the batch-corrected data.

## Karyotyping of organoids

Three days after splitting, colcemid was added to each well in self-renewing (SN) medium at 0.1 µg/ml for 48 hr. Organoids were then incubated in TrypLE™ Express Enzyme (ThermoFisher Scientific, 12604013) 37°C for 2 min and Advanced DMEM with 5% (v/v) foetal bovine serum (FBS) added. After centrifugation at 4°C, 300 g for 5 min, the cell pellet was resuspended in Advanced DMEM with 5% FBS and cells were triturated with a flame polished glass pipette. After further centrifugation at 1200 rpm for 7 min, 5 ml of 0.055 M KCl hypotonic solution was added to each tube which was gently inverted twice to mix. Tubes were centrifuged at 1200 rpm for 7 min. 500 µl of 3:1 100% methanol:glacial acetic acid fixative was added to each cell pellet dropwise down the side of the tube, then 1.5 ml was added in one go. Tubes were centrifuged at 1200 rpm for 7 min. The pellet was resuspended in fixative and stored at −20°C. All fixed cells were delivered to the Cytogenetics Laboratory at Cambridge University Hospitals NHS Foundation Trust for karyotyping.

## Xenotransplantation (Mouse kidney capsule)

E13.5 whole mouse lungs were microdissected from *Rosa26R-mT/mG* heterozygous embryos (10 lungs) and from MF1 embryos (24 lungs) and cut into small pieces. Lungs were incubated in TrypLE™ Express Enzyme (ThermoFisher Scientific, 12604013) at 37°C for 2 min then 20 ml of Advanced DMEM with 5% FBS added. After centrifugation at 4°C, 300 g for 5 min and aspiration, the cell pellet was resuspended in Advanced DMEM supplemented with 0.5% (v/v) BSA and cells were triturated with a flame polished glass pipette (no cell strainer was used). Cells were counted manually. Three biological replicates of human embryonic lung organoids were selected (~120 wells in total) and processed as the whole mouse lungs. Human and mouse cells were then combined at a ratio of 1:4 (human to mouse) with 2 million cells in total per kidney capsule graft. For each of the three biological replicates, three human cell samples were mixed with MF1 lungs and one with *Rosa26R-mT/mG/+* lungs. In order to facilitate kidney capsule transplantation, the human/mouse cell mixture was prepared into a cell aggregate (*Sheridan et al., 2009*). Cells were spun at 300 g for 5 min and 100 µl of the cell suspension was aspirated with a sterile non-filtered P200 tip, the end part of which was shortened slightly. The end of the tip was then pushed into folded parafilm and secured by combining all folds upwards (to prevent the cell suspension from leaking) and then the

whole tip was centrifuged in a 15 ml tube at 300 g for 5 min. A cell pellet became visible on the end part of the pipette nearest to the parafilm seal. The pipette tip was held horizontally to remove the parafilm and then the cell pellet was transferred directly onto a polycarbonate filter (Millipore) floating in a well of a 24 well plate. Overnight culture in self-renewing medium supplemented with 5% FBS was performed. The next morning, cell aggregates were transferred to Advanced DMEM. NSG male mice were anaesthetised and each cell transplanted into the left kidney capsule. Kidneys were harvested at 3, 7 and 12 weeks post-transplant and fixed overnight 4% PFA at 4°C. Chimeric human-mouse MF1 grafts were harvested at 3 and 7 weeks and the human cells distinguished using HuNu antibody staining. Human–mouse *Rosa26R-mT/mG* grafts were harvested at 12 weeks and the human cells distinguished by the absence of red membranes. The 12 week hosts received three daily intraperitoneal injections of 0.5 mg/Kg body weight dexamethasone 1 week before culling.

### Xenotransplantation (injured mouse lungs)

NSG male mice were used. Mice were anaesthetised using 2% (v/v) isoflurane and then exposed to bleomycin oropharyngeally through controlled aspiration on day −2. Bleomycin (clinical grade purchased from UCL pharmacy) was prepared as a 1 mg/ml stock using sterile 0.9% (w/v) normal saline and administered as 1 µl/g body weight to each mouse. Human embryonic lung organoids were expanded as described above. On the day of transplantation organoids were harvested in 15 ml tubes using cold washing medium, centrifuged at 4°C, 300 g for 5 min and incubated in TrypLE$^{TM}$ Express Enzyme (ThermoFisher Scientific, 12604013) at 37°C for 2 min then 20 ml of Advanced DMEM with 5% (v/v) FBS added. After centrifugation at 4°C, 300 g for 5 min and aspiration, the cell pellet was resuspended in DMEM/F12 supplemented with 0.5% (v/v) BSA and cells were triturated with a flame polished glass pipette. Viable cells were counted manually and single cell morphology confirmed. 600,000 cells (in 25 µl of DMEM/F12 supplemented with 0.5% (v/v) BSA) were administered intratracheally to each mouse under isoflurane anaesthesia on day 0. Lungs were harvested by culling the animals through intraperitoneal injection of sodium thiopental. Lungs were insufflated with 4% (w/v) PFA intratracheally, upon which the most proximal part of the trachea was tied with dental floss. The whole lung was then immersed in 4% (w/v) PFA and incubated overnight at 4°C. Tissue was then processed for cryo-sectioning.

### Microscopy

The following microscopes were used: Compound microscope: Zeiss Axiophot. Confocal microscope: 1) Leica SP8, 2) Olympus FV1000 Inverted. Confocal z stacks were acquired at an optical resolution of 1024 × 1024 with an optical z slice every 1 µm for 40x images and every 2.3 µm for 20x images. Movies of growing organoids were captured by culturing in a Nikon Biostation and capturing bright-field images every 12 hr for up to 11 days.

### Image scoring

For estimating the proportion of mesenchyme after microdissection of the epithelium for RNAseq and organoid culture, a macro for Fiji was written by Richard Butler, Gurdon Institute Imaging Facility (Mesenchyme_Macro.ijm is available as a supplemental file). The macro estimates the number of mesenchymal cells inside the 3D projection of a selected 2D region of interest by subtracting an E-Cadherin signal mask from a DAPI signal mask and dividing the remaining volume by a user-defined predicted nucleus volume.

Quantitation of number of human grafts seen in bleomycin-injured mouse lungs was performed manually by counting the number of grafts, and number of cells per graft, seen per 20 consecutive 20x fields in 1 section of 1 random slide for each of the four mouse lungs in each of the four experimental groups. Quantitation of the number of human cells per graft was also manual.

Quantitation of organoid forming efficiency with, or without, TGF$\beta$ inhibition was done based on the definition of an organoid as structures which had at least doubled in size compared to a fresh tip and had also branched. Quantitation of organoid size (diameter at the widest point) with, or without, TGF$\beta$ inhibition was expressed as a percentage increase comparing Day 11 to Day 1.

## Acknowledgements

Thanks to Maike Paramour for the RNAseq library construction. Charles Bradshaw for mapping RNA-seq reads. Charles-Etienne Dumeau for kidney capsule injections. Nick Bredenkamp and Maria Rostovskaya for assistance with cell pellet preparation for kidney capsule injections. Roger Barker for assistance with ethical approval and obtaining tissue. Leland Dobbs and Robert Gonzales for the kind gift of the HTII-280 and HT1-56 antibodies.

This study was supported by the Medical Research Council (G0900424 and MR/P009581/1, ELR), Wellcome Trust PhD programme for Clinicians (MZN), Addenbrooke's Charitable Trust (MZN), Fundació Universitària Agustí Pedro i Pons Short-term Fellowship (OC), Wellcome Trust clinical PhD fellowship (JAJ), EBPOD EMBL-EBI/Cambridge Biomedical Research Centre Postdoctoral Fellowship (KJH), IKER Basque postdoctoral fellowship (UL), COST BM1201. Core grants: Gurdon Institute: Wellcome Trust (092096), Cancer Research UK (C6946/A14492); Stem Cell Initiative: Wellcome Trust/MRC. The authors have no conflicts of interest to declare.

## Additional information

### Funding

| Funder | Grant reference number | Author |
| --- | --- | --- |
| Medical Research Council | G0900424 | Emma L Rawlins |
| Wellcome | PhD programme for clinicians fellowship | Marko Z Nikolić |
| Addenbrooke's Charitable Trust, Cambridge University Hospitals | | Marko Z Nikolić |
| Ikerbasque, Basque Foundation for Science | | Usua Laresgoiti |
| Cancer Research UK | C6946/A14492 | Emma L Rawlins |
| Medical Research Council | MR/P009581/1 | Emma L Rawlins |
| Wellcome | Clinical PhD Fellowship | Jo-Anne Johnson |
| Wellcome | 092096 | Emma L Rawlins |
| Fundació Universitària Agustí Pedro i Pons | Short term fellowship | Oriol Caritg |

The funders had no role in study design, data collection and interpretation, or the decision to submit the work for publication.

### Author contributions

MZN, Conceptualization, Funding acquisition, Investigation, Writing—review and editing, Writing - original draft, Visualisation, Methodology; OC, QJ, J-AJ, DS, JLB, Investigation; KJH, Formal analysis, Writing—review and editing; UL, Supervision; GA, Formal analysis; RB, Software, Formal analysis; MZ, Supervision, Methodology; AG, Investigation, Methodology; ELR, Conceptualization, Supervision, Funding acquisition, Investigation, Visualization, Methodology, Writing—original draft, Project administration, Writing—review and editing

### Author ORCIDs

Marko Z Nikolić, http://orcid.org/0000-0001-6304-6848
Emma L Rawlins, http://orcid.org/0000-0001-7426-3792

### Ethics

Human subjects: Human embryonic and foetal lung tissue. Human embryonic and foetal lungs were obtained from terminations of pregnancy from Cambridge University Hospitals NHS Foundation Trust under permission from NHS Research Ethical Committee (96/085) and the Joint MRC/Wellcome Trust Human Developmental Biology Resource (London and Newcastle, grant 099175/Z/12/Z, www.hdbr.org). Informed consent and consent to publish was obtained. Human adult lung tissue.

Fresh healthy adult lung tissue (background tissue from lobectomies for lung cancer) was obtained from Papworth Hospital NHS Foundation Trust (Research Tissue Bank Generic REC approval, Tissue Bank Project number T01939). Informed consent and consent to publish was obtained.

Animal experimentation: This research has been regulated under the Animals (Scientific Procedures) Act 1986 Amendment Regulations 2012 following ethical review by the University of Cambridge and University College London Animal Welfare and Ethical Review Body (AWERB). All experiments were thereafter conducted according to Home Office project licenses PPL 70/8012 (Emma Rawlins, University of Cambridge) and 70/7607 (Adam Giangreco, UCL).

## Additional files

### Supplementary files

• Supplementary file 1. Transcription factors and signalling molecules differentially expressed in human tip versus stalk (related to *Figure 2F–L*).

• Supplementary file 2. Transcription Factors with mouse orthologues upregulated > 2 fold in human tips or stalks: evidence for mouse expression pattern (related to *Figure 2J and L*).

### Major datasets

The following dataset was generated:

| Author(s) | Year | Dataset title | Dataset URL | Database, license, and accessibility information |
|---|---|---|---|---|
| Nikolić MZ, Johnson J, Howell KJ, Allen G, Zilbauer M, Rawlins EL | 2017 | Expression data from fresh human embryonic lung epithelial tip and stalk cells and cultured organoids derived from tip and stalk | https://www.ncbi.nlm.nih.gov/geo/query/acc.cgi?acc=GSE95860 | Publicly available at the NCBI Gene Expression Omnibus (accession no: GSE95860) |

The following previously published datasets were used:

| Author(s) | Year | Dataset title | Dataset URL | Database, license, and accessibility information |
|---|---|---|---|---|
| Laresgoiti U, Nikolić MZ, Rao C, Brady JL, et al | 2016 | Expression data from mouse embryonic lung epithelial tip progenitor cells | https://www.ncbi.nlm.nih.gov/geo/query/acc.cgi?acc=GSM1968996 | Publicly available at the NCBI Gene Expression Omnibus (accession no: GSM1968996) |
| Laresgoiti U, Nikolić MZ, Rao C, Brady JL, et al | 2016 | Microdissected E11.5 lung epithelial tip biological replicate 2 | https://www.ncbi.nlm.nih.gov/geo/query/acc.cgi?acc=GSM1968997 | Publicly available at the NCBI Gene Expression Omnibus (accession no: GSM1968997) |
| Laresgoiti U, Nikolić MZ, Rao C, Brady JL, et al | 2016 | Microdissected E11.5 lung epithelial tip biological replicate 3 | https://www.ncbi.nlm.nih.gov/geo/query/acc.cgi?acc=GSM1968998 | Publicly available at the NCBI Gene Expression Omnibus (accession no: GSM1968998) |
| Laresgoiti U, Nikolić MZ, Rao C, Brady JL, et al | 2016 | Microdissected E11.5 lung epithelial tip biological replicate 4 | https://www.ncbi.nlm.nih.gov/geo/query/acc.cgi?acc=GSM1968999 | Publicly available at the NCBI Gene Expression Omnibus (accession no: GSM1968999) |
| Laresgoiti U, Nikolić MZ, Rao C, Brady JL, et al | 2016 | Microdissected E11.5 lung epithelial tip biological replicate 5 | https://www.ncbi.nlm.nih.gov/geo/query/acc.cgi?acc=GSM1969000 | Publicly available at the NCBI Gene Expression Omnibus (accession no: GSM1969000) |

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
