## [Decision Letter]

Thank you for submitting your article "Human lung epithelial tips are multipotent progenitors that can be expanded as long-term self-renewing organoids" for consideration by *eLife*. Your article has been reviewed by three peer reviewers, and the evaluation has been overseen by Marianne Bronner as the Senior Editor and Reviewing Editor. The reviewers have opted to remain anonymous.

The reviewers have discussed the reviews with one another and the Reviewing Editor has drafted this decision to help you prepare a revised submission.

Summary:

Nikolic et al. present an impressive atlas of human lung development from 7-20 weeks of gestation, carefully mapping coexpression of canonical markers and generating gene expression datasets of microdissected distal and proximal lung regions. Strikingly, the authors demonstrate an important difference between human and mouse lung development: human distal tip progenitors in vivo express high levels of *SOX9* and low levels of *SOX2*, whereas in mouse, the cognate progenitors exclusively express *SOX9* (but not *SOX2*). They go on to demonstrate that human distal lung progenitors can be expanded in the presence of combined FGF agonists, EGF agonists, WNT agonists, a BMP inhibitor and a TGFb inhibitor for 4-9 months, during which they can be continually passaged and expanded. The authors' ability to access, document and expand human fetal lung progenitors is a major step forward for the field. The only real limitation is that the authors do not show that their expanded human fetal lung progenitors can engraft in vivo and robustly both airway and alveolar cell-types, and the in vitro differentiation into these lineages is also somewhat limited-however, the paper is still very impressive and will be of great interest to the developmental biology and stem cell communities even in its current state. The evidence for differentiation is robust and multiple markers are used, although the quality of the immunofluorescence could, in key cases, be improved.

Essential revisions:

1) The authors describe conditions in which undifferentiated epithelial cells isolated from the distal tips of human fetal lung can be cultured in vitro in 3D and efficiently expanded by self-renewal, while retaining a normal karyotype and ability to differentiate into both proximal airway and distal alveolar cell types. Importantly, the authors describe two transplantation assays: grafting into the bleomycin injured lung and the kidney capsule of immunocompromised mice. The evidence that human cells grow and form epithelial structures in the injured lung is good but it is hard to find definitive information about whether the cells express markers for mature cell types. For example, did the tall columnar cells in Figure 5, panel 8 express Foxj1 or mucins? What happens to the cells over the long term? How do these results compare with those reported by Rosen et al. Nature Medicine 21: 869-79 (2015)?

2) The evidence for differentiation, when combined with lung mesenchyme in vitro, of human 19 pcw lung tip cells into alveolar Type 1 and Type 2 cells, as well as cells that co-express markers for both cell types, is convincing. However, a wider panel of markers needs to be used to compare the putative bipotential progenitors seen in vitro with progenitor cells observed in vivo.

3) In vivo transplantation studies into bleomycin-injured mouse lungs are superficially analyzed. The authors examined only 2 and 8 days post-transplantation, while they show E-cadherin staining (Figure 5) of engrafted human cells, they claim in the text that *SOX9* and *SOX2* are still robustly coexpressed. This suggests that the grafts that formed were solely undifferentiated progenitors. Is NKX2.1 still expressed in the transplanted cells in bleomycin-injured lungs? That would be stronger proof that the cells have maintained a lung identity. Finally, given that these data are rather superficial, might be better suited for the supplementary figures.

4) The co-expression of *Sox2* and *Sox9* in distal tip cells is striking. This is clearly different between mouse and human cells. ID2 staining would be interesting.

5) The authors have employed a panel of markers to characterize the expression key genes in distal tip progenitors at various stages on human fetal development in Figure 1. The authors could show staining of Nkx2-1, FoxA2, and markers of AT1 cells such as AQP5 or RAGE.

6) The gene expression analysis in Figure 2 was well done with novel genes being discovered along the proximal-distal axis. The authors might check for protein expression of many of the fascinating TFs when antibody reagents are available. This might be powerful in identifying singular factors that mark distal tip progenitors.

7) The authors then establish an in vitro self-renewing culture conditions for human epithelial tips. This work was very well done and identified defined conditions that maintained *Sox2Sox2*^+^*Sox9*+ cells over long-term cultures (4-9 months). It would be useful to know if tip cultures can be frozen and thawed.

8) The global gene expression analysis indicates that the cultured tips have different gene expression signatures than freshly isolated ones. It would be useful to again establish TF expression at the protein level further characterizing these cells.

9) In Figure 6, the authors demonstrate the in vitro differentiation potential of tip organoids. We would like to see better characterization of alveolar differentiation using mesenchymal and AT1 markers post differentiation. Quantifying the reproducibility of differentiation would be helpful establish criterion for adequate cultures for the entire community.

---

## [Author Response]

*Essential revisions:*

*1) The authors describe conditions in which undifferentiated epithelial cells isolated from the distal tips of human fetal lung can be cultured in vitro in 3D and efficiently expanded by self-renewal, while retaining a normal karyotype and ability to differentiate into both proximal airway and distal alveolar cell types. Importantly, the authors describe two transplantation assays: grafting into the bleomycin injured lung and the kidney capsule of immunocompromised mice. The evidence that human cells grow and form epithelial structures in the injured lung is good but it is hard to find definitive information about whether the cells express markers for mature cell types. For example, did the tall columnar cells in Figure 5, panel 8 express Foxj1 or mucins? What happens to the cells over the long term? How do these results compare with those reported by Rosen et al. Nature Medicine 21: 869-79 (2015)?*

This comment is redundant with comment 3 and they are addressed together below.

*2) The evidence for differentiation, when combined with lung mesenchyme in vitro, of human 19 pcw lung tip cells into alveolar Type 1 and Type 2 cells, as well as cells that co-express markers for both cell types, is convincing. However, a wider panel of markers needs to be used to compare the putative bipotential progenitors seen in vitro with progenitor cells observed in vivo.*

In order to address this, we first had to validate additional cell fate markers for human embryonic lung alveolar epithelial development (see comment 5 also). We now provide detailed expression patterns for additional markers of mature AT2 and AT1 cell types and find, similar to our previous analysis of pro-SFTPC, HTII-280 and HOPX, that the additional markers are mostly co-expressed in bipotent progenitors and are not unique markers of cell fate at the stages of development that we have been able to study.

AT2 markers:

The only additional marker we have been able to obtain reliable, convincing staining for is ABCA3. This is expressed in mature AT2 cells in the adult lung, but cannot be detected at the stages of development we have studied (Figure 1—figure supplement 2). This is consistent with a previous report from the Whitsett lab (Stahlman, M.T., Besnard, V., Wert, S.E., Weaver, T.E., Dingle, S., Xu, Y., von Zychlin, K., Olson, S.J., Whitsett, J.A., 2007. Expression of ABCA3 in developing lung and other tissues. J Histochem Cytochem 55, 71-83).

AT1 markers:

PDPN is unique to AT1 cells in adult human lungs, but we show that is expressed at low levels in developing human lung epithelium from 9 pcw, is stronger in developing alveolar sacs where it co-localises with the adult AT2 marker HTII-280 at 17 pcw. Again, suggesting expression in bipotent progenitors (Figure 1—figure supplement 3).

AQP5 is unique to AT1 cells in adult human lungs. In mouse development it has been characterized as a marker of mature AT1 cells and as not expressed in the bipotent progenitors (Desai, T.J., Brownfield, D.G., Krasnow, M.A., 2014. Alveolar progenitor and stem cells in lung development, renewal and cancer. Nature 507, 190-194). In human embryonic lungs we find that AQP5 is also a bipotent progenitor marker, although it is expressed at higher levels in cells with cellular membrane extensions, likely differentiating AT1 cells (Figure 1—figure supplement 4).

At the suggestion of the reviewers, we also checked for heterogeneity of NKX2-1 and FOXA2 expression in the developing alveolar epithelium. However, at the stages of development that we can access, we found no evidence for variation in levels of these markers (Figure 1—figure supplement 5).

With this normal expression data in hand, we were able to go back and re-assess the alveolar differentiation state in the organoid-mesenchyme co-cultures. All results were consistent with our original conclusion that we have achieved differentiation to the bipotent progenitor stage, but not complete AT2/AT1 differentiation. These data are documented in Figure 6—figure supplement 4.

*3) In vivo transplantation studies into bleomycin-injured mouse lungs are superficially analyzed. The authors examined only 2 and 8 days post-transplantation, while they show E-cadherin staining (Figure 5) of engrafted human cells, they claim in the text that SOX9 and SOX2 are still robustly coexpressed. This suggests that the grafts that formed were solely undifferentiated progenitors. Is NKX2.1 still expressed in the transplanted cells in bleomycin-injured lungs? That would be stronger proof that the cells have maintained a lung identity. Finally, given that these data are rather superficial, might be better suited for the supplementary figures.*

As suggested by the reviewers we have performed further analysis of the in vivo transplantation studies. We now show in the figure that the grafts at day 8 post-transplant are *SOX2* and *SOX9* dual-positive. Interestingly, the grafted human cells also express FOXA2, but have mostly turned off NKX2-1. In spite of this, we observe that grafts have regions of KRT5 (airway basal fate), TRP63 (airway basal fate) and MUC5AC (goblet cell fate) positive cells. However, FOXJ1 and cilia were not observed which is maybe not surprising as the cells have only had 8 days to differentiate.

In summary, we are seeing the first signs of airway differentiation in the grafts, but no evidence of alveolar formation. Moreover, the grafted cells retain the progenitor markers *SOX9* and *SOX2*, but are switching off the lung epithelial marker NKX2-1. This is fully documented in Figure 5—figure supplement 1. Interestingly, we have previously observed that NKX2-1 is down-regulated if organoids are left to over-grow in self-renewing conditions without a media change. This suggests, that maintenance of NKX2-1 probably requires active signals which may not be present in the mouse lungs. The loss of NKX2-1 may also explain why we do not see alveolar-fated cells as NKX2-1 expression is required for distal lung development (Minoo, P., Su, G., Drum, H., Bringas, P., Kimura, S., 1999. Defects in tracheoesophageal and lung morphogenesis in Nkx2.1(-/-) mouse embryos. Dev Biol 209, 60-71).

In comment 1 it was suggested that we compare our grafting data with that of Rosen et al., 2015 who transplanted fresh human canalicular stage lung cells into injured mouse lungs by intra-venous infusion. It is very difficult to make a direct comparison. Our experiments used pseudoglandular stage tip epithelial cells that had been expanded as organoids, Nod Scid Gamma mice, bleomycin injury, intra-tracheal cell administration and analysis at day 2 and 8. By contrast, Rosen et al., used fresh dissociated canalicular stage human lungs, Nod Scid mice, naphthalene injury followed by irradiation, tail vein cell injections and analysis at 7 weeks post-cell administration. We therefore feel the experiments are too different to compare directly and have not commented on Rosen et al., within the text.

Finally, it was suggested by the reviewers that the grafting experiments are quite preliminary and should be moved to the supplemental data. We agree, further analysis would require longer-term experiments that are out-with the scope of this revision. Therefore, although we think they are informative for the reader and would like to include them, we have moved them to Figure 5—figure supplement 1.

*4) The co-expression of Sox2 and Sox9 in distal tip cells is striking. This is clearly different between mouse and human cells. ID2 staining would be interesting.*

We now include ID2 staining in Figure 2—figure supplement 4.

*5) The authors have employed a panel of markers to characterize the expression key genes in distal tip progenitors at various stages on human fetal development in Figure 1. The authors could show staining of Nkx2-1, FoxA2, and markers of AT1 cells such as AQP5 or RAGE.*

We have now performed these experiments as described in the answer to point 2 above. (Brief note: available RAGE antibodies did not work in our hands on human tissue. However, we were able to get PDPN antibodies working as an alternative.)

*6) The gene expression analysis in Figure 2 was well done with novel genes being discovered along the proximal-distal axis. The authors might check for protein expression of many of the fascinating TFs when antibody reagents are available. This might be powerful in identifying singular factors that mark distal tip progenitors.*

We attempted to detect protein expression for many of the transcription factors we identified and were able to get working antibodies against ETV5, HMGA1, HMGA2, HNF1B, ID2. These data are now presented in Figure 2—figure supplement 4.

*7) The authors then establish an in vitro self-renewing culture conditions for human epithelial tips. This work was very well done and identified defined conditions that maintained Sox2Sox2^+^Sox9+ cells over long-term cultures (4-9 months). It would be useful to know if tip cultures can be frozen and thawed.*

We freeze-thaw our organoids routinely and now mention this in the Discussion and include the protocol in the Materials and methods section.

*8) The global gene expression analysis indicates that the cultured tips have different gene expression signatures than freshly isolated ones. It would be useful to again establish TF expression at the protein level further characterizing these cells.*

This was a good suggestion and we were able to show that all of the transcription factors that we were able to detect at the tip in lung sections were also present in the self-renewing tip organoids: ETV5, HMGA1, HMGA2, HNF1B, NKX2-1, ID2, *SOX2, SOX9* (Figure 3—figure supplement 1).

*9) In Figure 6, the authors demonstrate the in vitro differentiation potential of tip organoids. We would like to see better characterization of alveolar differentiation using mesenchymal and AT1 markers post differentiation. Quantifying the reproducibility of differentiation would be helpful establish criterion for adequate cultures for the entire community.*

The epithelial component of this was covered in point 2 above. We have also assessed the mesenchyme that we are putting into the cultures. Note we looked directly at the expanded mesenchymal cells rather than at the mesenchyme at the end of the co-cultures when the phenotype of the mesenchymal cells could well have changed.

As for the epithelial component, we first had to validate the expression of mesenchymal markers in sections of human embryonic lungs at 20 pcw. Briefly, SMA surrounds the bronchioles and vessels and is present in a few alveolar mesenchymal cells also; PDGFRA is in a sub-set of mesenchymal cells sometimes overlapping with SMA; PDGFRB is ubiquitous in the 20 pcw lung mesenchyme; CD90 is located in scattered cells; VECAD is expressed in the endothelial cells (Figure 6—figure supplement 5).

The expanded mesenchyme was a heterogeneous population that did not express VECAD, as expected. It ubiquitously expressed PDGFRB. SMA; PDGFRA and CD90 were all heterogeneous within the expanded population (Figure 6—figure supplement 5).

We now show representative results from two independent organoid lines and two independent sets of expanded fibroblasts. Results from all technical (6x) and biological (2x) replicates have been identical.